# Analyzing & Reducing the Need for Learning Rate Warmup in GPT Training

**Atli Kosson**      **Bettina Messmer**      **Martin Jaggi**

EPFL, Switzerland

`firstname.lastname@epfl.ch`

## Abstract

Learning Rate Warmup is a popular heuristic for training neural networks, especially at larger batch sizes, despite limited understanding of its benefits. Warmup decreases the update size $\Delta \boldsymbol{w}_t = \eta_t \boldsymbol{u}_t$ early in training by using lower values for the learning rate $\eta_t$. In this work we argue that warmup benefits training by keeping the overall size of $\Delta \boldsymbol{w}_t$ limited, counteracting large initial values of $\boldsymbol{u}_t$. Focusing on small-scale GPT training with AdamW/Lion, we explore the following question: *Why and by which criteria are early updates $\boldsymbol{u}_t$ too large?* We analyze different metrics for the update size including the $\ell_2$-norm, resulting directional change, and impact on the representations of the network, providing a new perspective on warmup. In particular, we find that warmup helps counteract large angular updates as well as a limited critical batch size early in training. Finally, we show that the need for warmup can be significantly reduced or eliminated by modifying the optimizer to explicitly normalize $\boldsymbol{u}_t$ based on the aforementioned metrics.

## 1 Introduction

Neural networks are typically trained using variations of stochastic gradient descent. The weight updates $\Delta \boldsymbol{w}$ have the form $\Delta \boldsymbol{w} = \eta \boldsymbol{u}$, where $\eta$ denotes the *learning rate* and $\boldsymbol{u}$ an unscaled update vector derived from the history of weight gradients. Throughout training, the learning rate is often adjusted over time $t$ according to a *learning rate schedule*, $\eta = \eta_t$. This schedule frequently includes an initial phase known as a learning rate warmup, where the learning rate starts low and is increased to a target value before being reduced according to a decay schedule. Both the choice of warmup and decay strategy can significantly affect the final model performance. In this work, we focus on the linear warmup introduced by Goyal et al. [10] for large batch size ResNet [11] training, which is also commonly used for transformers [37].

The length of the warmup is a hyperparameter that requires tuning, which is complicated by the fact that the reasons for its effectiveness are somewhat unclear. Warmup empirically helps stabilize training and allows the use of larger learning rates throughout the rest of training, which can speed up the process and provide beneficial regularization [10]. Since the learning rate simply scales the size of the updates $\Delta \boldsymbol{w} = \eta \boldsymbol{u}$, warmup must achieve these effects by decreasing the size of early updates. However, it is not fully clear why this helps. *Are the initial updates too large for some reason?* For example, we might need small $\eta_t$ values to counteract large $\boldsymbol{u}$ values early in training. *How should we quantify what makes an update $\Delta \boldsymbol{w}$ large? Why do large updates adversely affect training?*

In this work, we explore warmup from this perspective, focusing on GPT2 [29] training with adaptive optimizers like AdamW [24] and Lion [3]. We identify three key issues that necessitate warmup:

1. The way Adam handles momentum can lead to artificially large initial updates $\Delta \boldsymbol{w}$.

2. Early updates $\Delta \boldsymbol{w}$ are large compared to the initial weight magnitude of $\boldsymbol{w}$ for matrices.

3. The gradients of early samples are highly correlated, limiting effective mini-batch sizes.

38th Conference on Neural Information Processing Systems (NeurIPS 2024).

We demonstrate that simple modifications to the optimizer can mitigate the first two issues: eliminating the momentum bias correction in AdamW and scaling matrix updates to match their magnitude, akin to the Rotational Optimizers by Kosson et al. [19]. We analyze the third issue in terms of the rate at which the internal neural representations of the network are changing (sometimes called feature learning). When the gradients of different samples are highly correlated these internal representations change too fast, which we conjecture can lead to issues with the non-linearities (e.g. activation functions) of the network. This can also be seen as the *critical batch size* [28] being too low early in training to enable the use of the peak learning rate. We derive a scaling factor based on the signal-to-noise ratio of the gradient that can help mitigate this, functioning like an automatic learning rate warmup. Alternatively, we show that using high momentum values in conjunction to the first two methods may suffice to enable efficient training without warmup.

## 2 Related Work

Learning rate warmups have been used since at least ResNet [11], where a lower constant learning rate was applied at the start of training. Earlier works may have employed similar concepts; for example, Sutskever et al. [35] utilized a momentum schedule that could induce a similar effect in the "effective learning rate" as defined by Fu et al. [6]. The practice of linear warmup in its current form was popularized by Goyal et al. [10] and Vaswani et al. [37].

Warmup has been studied indirectly in various neural network optimizer works. A notable example is RAdam [23], a modification of Adam [18] aimed at reducing the need for warmup. However, Ma and Yarats [26] demonstrated that RAdam essentially incorporates a fixed warmup schedule within the optimizer. In appx. C.1 we show that this warmup effect is insufficient in our setup. Relative optimizers like LARS [47] and LAMB [48] are also believed to reduce the necessity for warmup [21]. Bernstein et al. [2] propose a relative optimizer called Fromage and analyze how relative weight changes relate to representation changes, but differ from our approach in that they do not describe the effects of the gradient signal-to-noise ratio on this relationship. We build upon Kosson et al. [19] which showed that weight decay can make standard optimizers function as approximate relative optimizers and proposed variants that reduce the benefit of warmup without fully eliminating it.

The effect of warmup in transformers was empirically studied by Wortsman et al. [41]. Xiong et al. [43] proposed the pre-LN normalization placement for transformers, showing it reduces the need for warmup. Huang et al. [13] studied initialization in transformers showing a link to warmup.

Finally, warmup has been studied directly on its own. Gotmare et al. [9] studied the effect of warmup, finding it helps avoid overly large updates to the weights of later layers which could be frozen to achieve a similar benefit. Gilmer et al. [7] study the need for warmup from a curvature perspective, showing it may help "push" the optimization trajectory towards flatter regions where higher learning rates are stable. Smith et al. [32] arrive at a similar conclusion, there is a stable learning rate that varies throughout training based on the curvature which limits the learning rate early on, necessitating warmup. These works focus on SGD with momentum, but it is less clear how curvature affects Adam-like or relative optimizers as we discuss later.

The relation between stochastic gradient noise and learning rate has been studied in several works [46, 28, 49, 34, 22, 27]. They find that the update size can be increased roughly linearly with the batch size up to a certain *critical batch size* that depends on ratio of the mean and variance of the mini-batch gradient. We show how the signal-to-noise ratio (SNR) of the mini-batch gradient amplifies changes to the neural representations of a network given a normalized update in weight space. We observe that the SNR starts out high but decreases over time, which translates to large early changes in the internal representations without warmup.

## 3 Baseline Experimental Setup & Results

Our main experiments focus on the training of a 124M parameter GPT2 [29] model on the Open-WebText corpus [8]. The model has 12 transformer blocks with an embedding dimension of 768. Our base training is performed at batch size 480 with a sequence length of 1024. We train for 5000 iterations which translates into roughly 20 tokens per parameter, as suggested by Chinchila [12]. The baselines use AdamW [24] (see algo. 1) with weight decay $\lambda = 0.1$, momentum coefficient $\beta_1 = 0.9$, smoothing coefficient $\beta_2 = 0.95$, and $\varepsilon = 10^{-8}$. The learning rate schedule consists of a

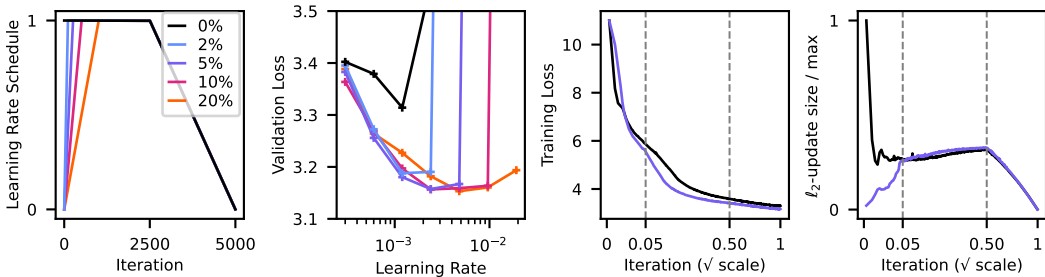

**Figure 1:** Warmup significantly benefits GPT2 training with AdamW. **Panel 1:** Trapezoidal learning rate schedules with different warmup lengths and 50% linear cooldown. **Panel 2:** Final validation loss for various learning rate and warmup configurations. Note the performance gap between no-warmup (black) and other configurations. **Panel 3:** Training curves comparing the best no-warmup run to a 5% warmup with the same learning rate. The warmup run quickly surpasses the no-warmup run. **Panel 4:** Comparison of $\ell_2$ update norms for these runs shows large initial updates without warmup.

linear warmup followed by a constant phase and eventually linear cooldown spanning half of training (see examples in fig. 1). This schedule keeps the peak learning rate and decay phase identical for different warmup lengths. This differs from other schedules, e.g. cosine, where the warmup length typically affects the whole shape. The learning rate value and the warmup length are swept for various configurations. Our code is based on NanoGPT [16] with additional utilities by Kosson et al. [19]. The hyperparameter values and base training configuration are adopted from NanoGPT. See appx. C.2 for experiments on additional architectures and datasets.

Figure 1 shows the baseline performance for our setup. We observe that even short warmup can significantly improve performance. Not using warmup results in faster initial progress for a given learning rate, but eventually falls behind leaving a permanent gap. Warmup not only stabilizes higher learning rates, but also prevents a lasting degradation of the model that can not be mitigated by simply training for slightly longer. We notice that although Adam normalizes the update size, its $\ell_2$-magnitude varies significantly throughout training with a large spike at the start of training.

## 4 The Interaction of Momentum and the $\ell_2$-Update Norm in AdamW

In this section we analyze the reason behind the large $\ell_2$-norm of early updates in our AdamW baseline seen in panel 4 of fig. 1. We find that this primarily stems from the $\beta_1$ bias correction. We then explore to what extent these large initial updates contribute to the need for warmup by modifying the optimizer to directly control the $\ell_2$-norm of the update. Although we find this is to be insufficient to replace warmup on its own, these changes are an important component of our later methods.

Adam-like optimizers such as AdamW (algo. 1) differ from simpler methods like SGD with momentum in that they normalize the update size with the gradient magnitude. This makes them invariant to a rescaling of the loss function and helps counteract potential differences in the gradient magnitude between layers. An important consequence of this is that the unscaled updates $\boldsymbol{u}$ are not large simply due to large initial gradients, unlike in plain SGD and other optimizers that don't normalize their

---

**Algorithm 1** AdamW (PyTorch variant, differing from the original by Loshchilov and Hutter [24])

**Require:** Learning rate $\eta_t$, weight decay $\lambda$, momentum $\beta_1$, magnitude smoothing $\beta_2$, $\varepsilon$ for numerical stability
1: **Initialize:** Time step $t \leftarrow 0$, parameter vector $\boldsymbol{\theta}_0$, momentum vector $\boldsymbol{m}_0 \leftarrow 0$, magnitude vector $\boldsymbol{v}_0 \leftarrow 0$
2: **while** stopping criteria not met **:**
3: $\quad$ $t \leftarrow t + 1$
4: $\quad$ $\boldsymbol{g}_t \leftarrow$ Mini-batch gradient w.r.t. $\boldsymbol{\theta}_{t-1}$
5: $\quad$ $\boldsymbol{m}_t \leftarrow \beta_1 \boldsymbol{m}_{t-1} + (1 - \beta_1)\boldsymbol{g}_t$
6: $\quad$ $\boldsymbol{v}_t \leftarrow \beta_2 \boldsymbol{v}_{t-1} + (1 - \beta_2)\boldsymbol{g}_t^2$
7: $\quad$ $\hat{\boldsymbol{m}}_t \leftarrow \boldsymbol{m}_t/(1 - \beta_1^t)$
8: $\quad$ $\hat{\boldsymbol{v}}_t \leftarrow \boldsymbol{v}_t/(1 - \beta_2^t)$
9: $\quad$ $\boldsymbol{\theta}_t \leftarrow (1 - \eta_t \lambda)\boldsymbol{\theta}_{t-1} - \eta_t \hat{m}_t/(\sqrt{\hat{v}_t} + \varepsilon)$

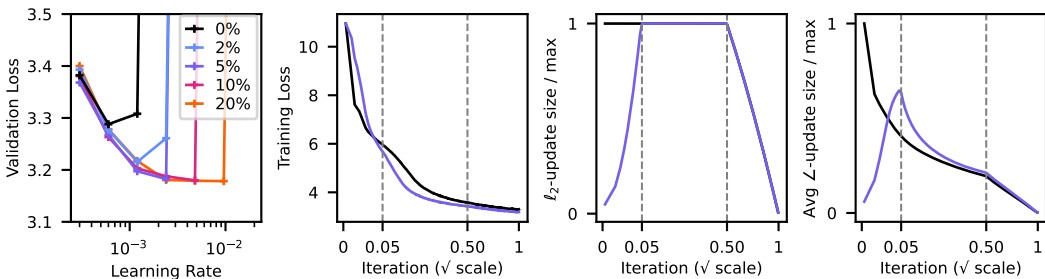

**Figure 2:** LionA (algo. 2) fails to significantly reduce the warmup advantage. **Panel 1:** Final validation loss across various learning rates and warmup percentages shows a reduced but still significant no-warmup penalty compared to AdamW (fig. 1). **Panel 2:** Training curves for 0% vs. 5% warmup at the highest stable learning rate for 0%, with warmup quickly overtaking no-warmup as before. **Panel 3:** LionA successfully controls the $\ell_2$-update norm. **Panel 4:** Early angular updates (see §5) are large without warmup and do not follow the learning rate schedule throughout training.

updates. Such un-normalized optimizers might diverge to infinity if a high learning rate is combined with large initial gradients or large curvature, as the update size is unbounded. Preventing this could be an additional benefit of warmup for SGD on top of the effects discussed in this work.

Although AdamW normalizes the update size based on the gradient, its magnitude can still vary throughout training as seen in fig. 1. This can be caused by changes in the gradient magnitude over time, especially when using different values of $\beta_1$ and $\beta_2$. However, it can also be caused by momentum and especially the bias correction (algo. 1, line 7). The magnitude of $m_t$ depends on the alignment of subsequent gradients $g_1, \ldots, g_t$ whereas the normalization factor $v_t$ does not. For example, when each $g_t$ is an independent zero-mean random vector with a fixed second moment $\mathbb{E}[g_t^2] = \sigma^2$, we have (see appx. B.1 for details):

$$\mathbb{E}[m_t^2] = (1 - \beta_1^{2t})\frac{1 - \beta_1}{1 + \beta_1}\sigma^2, \qquad \mathbb{E}[v_t] = (1 - \beta_2^t)\sigma^2 \tag{1}$$

In this case the bias correction for $\beta_1$ is incorrect since it is derived for a constant gradient. With the bias correction the size becomes $\mathbb{E}[\|\hat{m}\|^2] = \frac{1 + \beta_1^t}{1 - \beta_1^t}\frac{1 - \beta_1}{1 + \beta_1}\sigma^2$, amplifying the norm of early updates by $\sqrt{(1 + \beta_1^t)/(1 - \beta_1^t)}$. This factor is larger if the gradients between successive steps are negatively correlated, which we empirically observe happening in our setup (see §6.2).

The $\ell_2$-norm of AdamW updates can therefore vary significantly due to the initial bias correction, changes in the alignment of the gradients throughout training, and potential variations in the gradient norm over time. Lion [3] is a closely related optimizer that uses an element-wise sign operation to normalize the update, giving $+1$ for positive values, $-1$ for negative values and $0$ for zeros. Ignoring the possibility of zeros, this gives a constant update norm. Lion is closely related to Adam, and can be obtained by tracking the size of $m_t$ instead of $g_t$ in line 6 while setting $\beta_2 = 0$. It also uses a form of scaled Nesterov momentum instead of the traditional heavy-ball variant and a hyperparameter specification that differs significantly from AdamW. We propose a Lion variant, LionA (algo. 2), that keeps the hyperparameters more compatible with those of AdamW. The learning rate is kept comparable by scaling the $\ell_2$ update size to match that of AdamW in the random-gradient scenario, see appx. B.1 for the derivation of the scaling factors. Due to its ability to perfectly control the size of each update, we use Lion based methods for the remaining of this paper. This avoids confounding effects in Adam-like optimizers, such as $v$ being inaccurate from rapidly decreasing gradient magnitudes early in training, which can induce an additional warmup-like effect.

In fig. 2 we repeat the baseline sweep using LionA. **Despite perfect control of the $\ell_2$ update norm (as seen in panel 3), the benefit of warmup remains. This leads us to conclude that the $\ell_2$ update size is not sufficient to quantify the "effectively" large updates that we conjecture warmup mitigates.** The final panel shows that the angular update size (see definition in the following section), proposed to be a better measure of an effective step size by Wan et al. [38], still varies throughout training with a spike at the start of training. In the next section we explore the reasons for the large initial angular updates and how they significantly contribute to the need for warmup.

**Algorithm 2** LionA: A modified version of the Lion [3] optimizer for greater compatibility with AdamW (algo. 1). The sign operation replaces the magnitude smoothing, explicitly controlling the $\ell_2$-norm of each update. Additional scaling keeps the hyperparameters $\eta$, $\lambda$ comparable to AdamW.

---

**Require:** Learning rate $\eta_t$, weight decay $\lambda$, momentum $\beta$, Nesterov flag $\nu$
1: **Initialize:** Time step $t \leftarrow 0$, parameter vector $\boldsymbol{\theta}_0$, momentum vector $\boldsymbol{m}_0 \leftarrow 0$
2: **while** stopping criteria not met **:**
3: $\quad$ $t \leftarrow t + 1$
4: $\quad$ $\boldsymbol{g}_t \leftarrow$ Mini-batch gradient w.r.t. $\boldsymbol{\theta}_{t-1}$
5: $\quad$ $\boldsymbol{m}_t \leftarrow \beta \boldsymbol{m}_{t-1} + (1 - \beta) \boldsymbol{g}_t$
6: $\quad$ **if** Nesterov flag $\nu$ is set **:**
7: $\quad\quad$ $\boldsymbol{\theta}_t \leftarrow (1 - \eta_t \lambda) \boldsymbol{\theta}_{t-1} - \eta_t \cdot \sqrt{(1 - \beta^2)^2 + \beta^4 \frac{1-\beta}{1+\beta}} \cdot \mathrm{sign}(\beta \boldsymbol{m}_t + (1 - \beta) \boldsymbol{g}_t)$
8: $\quad$ **else:**
9: $\quad\quad$ $\boldsymbol{\theta}_t \leftarrow (1 - \eta_t \lambda) \boldsymbol{\theta}_{t-1} - \eta_t \cdot \sqrt{\frac{1-\beta}{1+\beta}} \cdot \mathrm{sign}(\boldsymbol{m}_t)$

---

## 5 The Importance and Irregularity of the Angular Update Size

The effect of a weight vector $\boldsymbol{w}_t \in \mathbb{R}^C$ used in a dot product with some vector $\boldsymbol{x}$ (e.g., in a neuron):

$$\langle \boldsymbol{w}_t, \boldsymbol{x} \rangle = \|\boldsymbol{w}_t\| \|\boldsymbol{x}\| \cos\left(\angle(\boldsymbol{w}_t, \boldsymbol{x})\right) \tag{2}$$

can be understood in terms of its magnitude $\|\boldsymbol{w}_t\|$ and direction $\boldsymbol{w}_t / \|\boldsymbol{w}_t\|$. The magnitude acts like a gain, scaling the outputs, whereas the direction determines which input representations $\boldsymbol{x}$ the system responds to. The angular update size [38] of an update $\boldsymbol{w}_t \mapsto \boldsymbol{w}_{t+1}$ is defined as:

$$\angle(\boldsymbol{w}_{t+1}, \boldsymbol{w}_t) = \arccos\left(\frac{\langle \boldsymbol{w}_{t-1}, \boldsymbol{w}_{t+1} \rangle}{\|\boldsymbol{w}_t\| \|\boldsymbol{w}_t\|}\right) \tag{3}$$

and measures how fast the direction of $\boldsymbol{w}_t$ changes during training, and thus its "preference" for $\boldsymbol{x}$.

With BatchNorm [15] and similar operations [1, 14, 42, 4], a network can become invariant to the magnitude of weight vectors like $\boldsymbol{w}_t$, such that only the direction matters and the vector is said to be *scale-invariant*. Weight Normalization [30] provides a good example of this, changing the system to:

$$\langle \boldsymbol{w}_t / \|\boldsymbol{w}_t\|, \boldsymbol{x} \rangle = \|\boldsymbol{x}\| \cos\left(\angle(\boldsymbol{w}_t, \boldsymbol{x})\right) \tag{4}$$

Note that although the system output is invariant to the magnitude $\|\boldsymbol{w}_t\|$, traditional optimizers are not. Scaling the value of a scale-invariant weight vector by a factor of $c > 0$, results in a gradient that is scaled by $c^{-1}$ and curvature that is scaled by $c^{-2}$ (see appx. B.2). For SGD this scales the angular update by $c^{-2}$ and for Adam-like optimizers it is scaled by $c^{-1}$. With weight decay the magnitude of scale-invariant vectors trends towards a certain stable equilibrium value over time which also results in a specific expected angular update size as described by Wan et al. [38], Kosson et al. [19].

This has several important implications. Changing the initialization magnitude of scale-invariant weights will scale the angular updates over time for standard optimizers, resulting in effects similar to modifying the learning rate schedule. For initial weight magnitudes that are small compared to the equilibrium magnitude, the early angular updates will be large and these optimizers may benefit from learning rate warmup to counteract this. These effects also make the notion of "curvature" somewhat arbitrary as it can be scaled without changing the encoded function. Optimizers that specifically account for the weight magnitude would be invariant to these effects which may reduce the need for warmup from the traditional curvature perspective. Although standard transformers are not fully scale-invariant, many of the angular update insights still approximately hold for un-normalized weights [19].

In light of this, we modify LionA to better control the angular update size by making the updates to weight matrices proportional to their weight magnitude, resulting in algo. 3. We normalize the angular update size to match the equilibrium value, replacing weight decay with projections similar to Kosson et al. [19]. However, unlike their RVs, we make the angular updates proportional to the learning rate schedule which we found was necessary for good performance in our case. We also do not rely on additional exponential moving averages to control the angular update size, instead utilizing the fixed update size from the LionA optimizer. This is similar to the Adam scheme used by Karras et al. [17] with good results for diffusion models. No additional normalization operations or scaling factors are introduced, which we still find to result in decent performance.

---

**Algorithm 3** LionAR: A rotational version of algo. 2 inspired by Kosson et al. [19]. The parameter vector is divided into sub-vectors $\boldsymbol{\theta} = [\boldsymbol{\theta}^{(1)}, \ldots, \boldsymbol{\theta}^{(P)}]$, each corresponding to either the weight vector of a neuron (e.g. a matrix row / a convolutional filter), or other parameters such as gains and biases. The updates of neuronal weight vectors are scaled to be proportional to their magnitude which is kept constant through projections that replace weight decay. Additional hyperparameter adjustments are made for compatibility with AdamW. The weight decay hyperparameter remains, fulfilling its primary role as a scaling factor for the relative updates of neurons [19].

---

**Require:** Learning rate $\eta_t$, weight decay $\lambda$, momentum $\beta$, Nesterov flag $\nu$
 1: **Initialize:** Time step $t \leftarrow 0$, parameter vector $\boldsymbol{\theta}_0$, momentum vector $\boldsymbol{m}_0 \leftarrow 0$
 2: **while** stopping criteria not met **:**
 3:     $t \leftarrow t + 1$
 4:     $[\boldsymbol{g}_t^{(1)}, \ldots, \boldsymbol{g}_t^{(P)}] \leftarrow$ Mini-batch gradient w.r.t. $\boldsymbol{\theta}_{t-1}$, divided into sub-vectors like $\boldsymbol{\theta}$
 5:     **for** $p \in \{1, \ldots, P\}$ **:**
 6:        $\boldsymbol{m}_t^{(p)} \leftarrow \beta \boldsymbol{m}_{t-1}^{(p)} + (1-\beta) \boldsymbol{g}_t^{(p)}$
 7:        **if** Nesterov flag $\nu$ is set **:**
 8:           $\boldsymbol{u}_t^{(p)} \leftarrow \beta \boldsymbol{m}_t^{(p)} + (1-\beta) \boldsymbol{g}_t^{(p)}$
 9:           $\gamma \leftarrow \sqrt{(1-\beta^2)^2 + \beta^4 \frac{1-\beta}{1+\beta}}$          *# Nesterov momentum scaling factor*
10:        **else:**
11:           $\boldsymbol{u}_t^{(p)} \leftarrow \boldsymbol{m}_t^{(p)}$
12:           $\gamma \leftarrow \sqrt{\frac{1-\beta}{1+\beta}}$          *# Heavy-ball momentum scaling factor*
13:        **if** $\boldsymbol{\theta}^{(p)} \in \mathbb{R}^C$ is a neuronal weight vector **:**
14:           $\hat{\boldsymbol{\theta}}_t^{(p)} \leftarrow \boldsymbol{\theta}_{t-1}^{(p)} - \frac{\eta_t}{\max_\tau(\eta_\tau)} \cdot \sqrt{2\max_\tau(\eta_\tau)\lambda} \cdot \gamma \cdot (\|\boldsymbol{\theta}_0^{(p)}\|/\sqrt{C}) \cdot \mathrm{sign}(\boldsymbol{u}_t^{(p)})$
15:           $\boldsymbol{\theta}_t^{(p)} \leftarrow \hat{\boldsymbol{\theta}}_t^{(p)} \cdot \|\boldsymbol{\theta}_0^{(p)}\|/\|\hat{\boldsymbol{\theta}}_t^{(p)}\|$          *# Reset the magnitude to the initial value*
16:        **else:**
17:           $\boldsymbol{\theta}_t^{(p)} \leftarrow \boldsymbol{\theta}_{t-1}^{(p)} - \eta_t \cdot \gamma \cdot \mathrm{sign}(\boldsymbol{u}_t^{(p)})$

---

**Figure 3:** LionAR (algo. 3) reduces but does not fully eliminate the benefit of warmup. **Panel 1:** LionAR is more stable across learning rates and shows a reduced but still significant performance gap without warmup. **Panel 2:** Comparing the 0% and 5% warmup for learning rate $\approx 10^{-2}$ shows the warmup run overtaking early in training. **Panel 3:** LionAR precisely controls the angular update size throughout training. **Panel 4:** Despite fixed angular (and thus relative) updates in weight space, the relative change of the internal representations (see §6) is large initially without warmup.

Figure 3 repeats the GPT2 training sweep with LionAR. Consistent with the findings of Kosson et al. [19] **we find that controlling the angular updates stabilizes training and decreases the benefit from warmup, but does not entirely eliminate it in this setting**. Both the angular change and the $\ell_2$-norm are simple measures of the update magnitude in parameter space that do not account for the direction or other aspects of the update. In the next section we show how a fixed update size in parameter space can result in large changes to the internal representations of the network (a.k.a. features, activations etc), as shown in the final panel of fig. 3.

## 6    Early Gradient Alignment Results in Large Representation Changes

Our two approaches to measuring and controlling the update size in weight space failed to fully explain the need for warmup. As an alternative to the parameters, we can analyze changes in

the internal representations or activations of the neural network (feature learning). Although this is harder to analyze and control, it may ultimately be a better measure of the true impact of an update. A parameter update can only affect the network output, and hence the loss, by changing the representation of the network inputs at some layer. Large, sudden, changes in the representations could significantly affect the non-linearities, potentially causing lasting issues such as dead ReLUs or vanishing gradients from saturated sigmoids. This could in turn explain the lasting performance degradation observed without warmup.

A given parameter update will affect the representations of each distinct input sample differently. The gradients computed on these samples also generally differ, but can align to some extent. For a higher gradient alignment, the impact of a parameter update of a given magnitude on the representations will be larger than otherwise. We will analyze this for a dot product making up a single neuron:

$$\boldsymbol{y} = \boldsymbol{w}^\top \boldsymbol{X} = [y_1, \ldots, y_B]^\top = [\langle \boldsymbol{w}, \boldsymbol{x}_1 \rangle, \ldots, \langle \boldsymbol{w}, \boldsymbol{x}_B \rangle]^\top \tag{5}$$

where $\boldsymbol{X} = [\boldsymbol{x}_1, \ldots, \boldsymbol{x}_B] \in \mathbb{R}^{C \times B}$ are the $C$-dimensional representations of a random mini-batch of $B$ inputs that is fed into the neuron, $\boldsymbol{w} \in \mathbb{R}^C$ is the weight vector, and $\boldsymbol{y} \in \mathbb{R}^B$ is a batch of outputs. For a weight update $\boldsymbol{w} \mapsto \boldsymbol{w} + \Delta \boldsymbol{w}$, we aim to quantify the size of the output change $\Delta \boldsymbol{y} = \Delta \boldsymbol{w}^\top \boldsymbol{X}$ computed on the same inputs. We focus on the *Relative Representation Change (RRC)*:

$$\frac{\|\Delta \boldsymbol{y}\|}{\|\boldsymbol{y}\|} = \frac{\|\Delta \boldsymbol{w}^\top \boldsymbol{X}\|}{\|\boldsymbol{w}^\top \boldsymbol{X}\|} \tag{6}$$

similar to the angular weight updates, as the sensitivity to the absolute change $\|\Delta \boldsymbol{y}\|$ can be unclear due to normalization or other scaling operations. Note that this is a measure of a *local change*, not accounting for changes in the inputs $\boldsymbol{X}$ from updates to preceding layers (*global change*).

## 6.1 Normalized Gradient Descent

We will focus our analysis on the relatively tractable case of normalized gradient descent with updates:

$$\Delta \boldsymbol{w} = -\eta \frac{\boldsymbol{g}}{\sqrt{\mathbb{E}[\|\boldsymbol{g}\|^2]}}, \qquad \boldsymbol{g} = \frac{1}{B} \sum_{b=1}^{B} \boldsymbol{g}_b \tag{7}$$

where $\boldsymbol{g}_b$ is the gradient of some loss w.r.t. $\boldsymbol{w}$ for the $b$-th element of the mini-batch. We will use the following definitions, properties, lemmas and assumptions for this system (see appx. B.4 for details):

- D1: We define $\boldsymbol{g}_b =: \bar{\boldsymbol{g}} + \tilde{\boldsymbol{g}}_b$ where $\bar{\boldsymbol{g}} = \mathbb{E}[\boldsymbol{g}]$ and $\tilde{\boldsymbol{g}}_b$ is the difference with $\mathbb{E}[\tilde{\boldsymbol{g}}_b] = \boldsymbol{0}$.
- D2: We define $\varphi := \mathbb{E}[\|\bar{\boldsymbol{g}}\|^2]/\mathbb{E}[\|\tilde{\boldsymbol{g}}_b\|^2]$ as the Signal-to-Noise Ratio (SNR) of the gradient.
- P1: For a neuron, $\boldsymbol{g}_b \parallel \boldsymbol{x}_b$, and hence $\boldsymbol{x}_b = \text{sign}(\langle \boldsymbol{x}_b, \boldsymbol{g}_b \rangle) \cdot (\|\boldsymbol{x}_b\|/\|\boldsymbol{g}_b\|) \cdot (\bar{\boldsymbol{g}} + \tilde{\boldsymbol{g}}_b)$.
- L1: Consider two independent random vectors $\boldsymbol{a} \in \mathbb{R}^C$ and $\boldsymbol{b} \in \mathbb{R}^C$, whose elements are independent and identically distributed (IID). If at least one of the vectors has a zero-mean distribution, then the expected value of the squared inner product of $\boldsymbol{a}$ and $\boldsymbol{b}$ is given by $\mathbb{E}[\langle \boldsymbol{a}, \boldsymbol{b} \rangle^2] = \mathbb{E}[\|\boldsymbol{a}\|^2]\mathbb{E}[\|\boldsymbol{b}\|^2]/C$.
- A1: We assume the following vector pairs satisfy L1: $(\boldsymbol{x}_i, \tilde{\boldsymbol{g}}_b)$ when $i \neq b$, $(\bar{\boldsymbol{g}}, \tilde{\boldsymbol{g}}_b)$ and $(\boldsymbol{w}, \boldsymbol{x}_b)$.

This allows us to compute the expected square relative representation change (detailed in appx. B.4):

$$\frac{\mathbb{E}[(\Delta y_b)^2]}{\mathbb{E}[y_b^2]} = \frac{\eta^2 C}{B^2 \|\boldsymbol{w}\|^2} \frac{1}{\mathbb{E}[\|\boldsymbol{g}\|^2]} \left( \mathbb{E}[\|\boldsymbol{g}_b\|^2] + \frac{B-1}{C} \mathbb{E}[\|\tilde{\boldsymbol{g}}_i\|^2] \right.$$

$$\left. + \frac{(B-1)^2}{\mathbb{E}[\|\boldsymbol{g}_b\|^2]} \left( \|\bar{\boldsymbol{g}}\|^4 + \frac{\|\bar{\boldsymbol{g}}\|^2 \mathbb{E}[\|\tilde{\boldsymbol{g}}_b\|^2]}{C} \right) + 2(B-1)\|\bar{\boldsymbol{g}}\|^2 \right) \tag{8}$$

$$= \frac{\eta^2 C}{B^2 \|\boldsymbol{w}\|^2} \frac{1}{\varphi + \frac{1}{B}} \left( (\varphi+1) + \frac{B-1}{C} + \left( \frac{(B-1)^2 \varphi}{\varphi+1} \left( \varphi + \frac{1}{C} \right) + 2(B-1)\varphi \right) \right) \tag{9}$$

The expected relative change in the output for a given sample can be broken down into three sources, the contribution of the sample itself (first term), random interference from the "noise" $\tilde{\boldsymbol{g}}_i$ of other samples (second term), and finally amplification of the common mean component $\bar{\boldsymbol{g}}$ (third term).

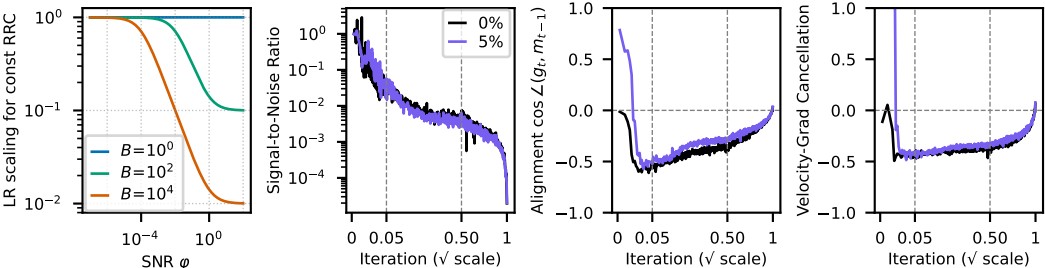

**Figure 4:** Equation (9) predicts that the learning rate needs to be downscaled for higher signal to noise ratios ($\varphi$) to keep the relative representation change constant. Larger batch sizes are affected more, with scaling becoming significant when $\varphi > B^{-1}$. **Panel 2:** Measurements of the SNR for the two highlighted runs in fig. 3. Note the SNR starts very high but is also remains large in comparison to our $B = 480$ for almost all of training. **Panel 3:** The gradient is strongly oppositely aligned with the momentum vector for most of training (shown for an example layer). **Panel 4:** Projecting the momentum component of the updates onto the gradient component shows that this results in the momentum vector "cancelling" roughly half the gradient on average.

The RRC expression provides many interesting insights. In the case of large input dimension $C \to \infty$ and small SNR $\varphi \approx 0$, keeping the RRC constant for different batch sizes involves scaling the learning rate $\eta \propto \sqrt{B}$, as suggested by Malladi et al. [27] for Adam. When the SNR $\varphi$ is some finite and value and $C$ is still large, this scaling rule instead starts to break down around $B = 1/\varphi$, matching the predicted critical batch size of e.g. McCandlish et al. [28]. The role of the dimension $C$ in the expression is curious, suggesting that narrower layers experience larger changes due to random inference from other samples in a given batch. The $C$ in the leading factor also suggests that the angular updates can be smaller for a larger input dimension, similar to what is proposed in $\mu$-parameterization [44, 45]. Most importantly, **this expression shows that if the SNR changes throughout training the learning rate needs to be adjusted to keep the RRC constant. In particular, with large batch sizes, a high initial SNR results in large representation changes which warmup can help prevent.** The first panel of fig. 4 shows how eq. (9) predicts we should downscale the learning rate for different batch sizes and SNRs, assuming we originally scaled the learning rate $\eta \propto \sqrt{B}$ and that $C$ is large. The second panel confirms that the SNR indeed starts out large, suggesting lower initial learning rates are needed, i.e. warmup.

In the first panel of fig. 5, we show the results of adding a term that scales the update size as predicted by eq. (9). **This acts similar to an automatic warmup based on online measurements of the SNR which we obtain from the gradient accumulation of micro-batches.** Although this helps close the gap between warmup and no-warmup, the overall performance is slightly worse. One potential issue is that our batch size of 480 is quite large compared to the measured SNR, exceeding the critical batch size estimation throughout most of training. This results in a scaling of the step size throughout training, which distorts the decay phase. It also requires large learning rate values to counteract the scaling, which may destabilize the training of non-matrix weights like gains. We increase the weight decay by a factor of $32\times$ to try to increase the angular updates relative to gains in order to compensate, but this value was not tuned and is unlikely to be optimal. We believe approaches that aim to directly control the RRC are a promising direction but require further work to be practical.

## 6.2 The Role of Momentum

Momentum is believed to be a key enabler of optimization with larger batch sizes [32, 39, 31]. However, it is unclear how it should change predictions for the critical batch size or relative representation change. Momentum spreads the application of a gradient out over multiple steps which tends to make each update smaller, especially for a random walk, which is reflected in our scaling coefficients in algo. 2 and 3. The smaller updates are counteracted by an increased correlation in their direction, which can result in similar "long term" changes from each gradient sample, especially for simpler methods like SGD that don't normalize the step size. In the last two panels of fig. 4 we observe that in our setup the gradient and momentum are negatively correlated, counteracting each other. We find momentum crucial for performant training, panel 2 of fig. 5 shows significant degradation without it.

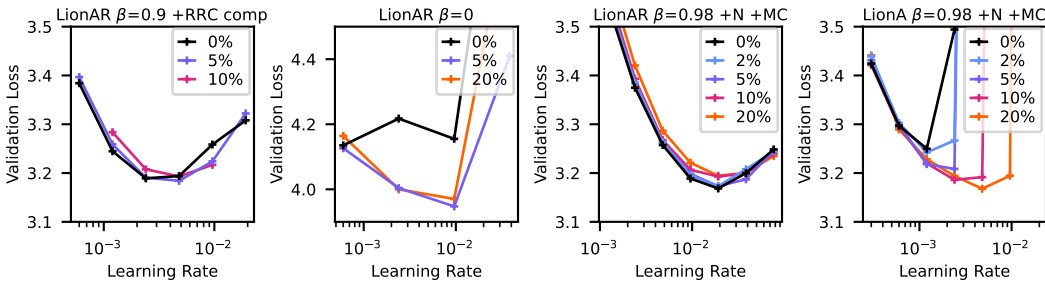

**Figure 5: Panel 1:** LionAR with a correction factor for the RRC based on eq. (9) does not benefit from a warmup. **Panel 2:** LionAR training without momentum results in drastically lower performance. **Panel 3:** In LionAR with increased momentum $\beta = 0.98$, Nesterov momentum and an inverse bias correction for early momentum, no warmup performs best. **Panel 4:** The same does not apply to LionA, suggesting that these changes are not sufficient without controlling the angular updates.

We believe the smaller update sizes for momentum combined with the potential for later gradients to counteract earlier gradients during their application over time, can help stabilize training. An otherwise large relative representation change is spread out over multiple steps and counteracted by later gradients. Higher values of momentum should amplify these effects. Looking at the total contribution of each gradient also implies that **with momentum early updates should be smaller when measured in parameter space, otherwise the relative representation change for those samples is too large.** This is equivalent to entirely removing the $\beta_1$ bias correction in AdamW (algo. 1, line 7), or introducing *an inverse bias correction* in Lion like algorithms (see appx. B.1 for details). Higher $\beta$ values should help amplify the stabilization effects of momentum. **In fig. 5 we find that at higher momentum values LionAR no longer benefits from warmup unlike LionA which still needs it**. These experiments use Nesterov momentum and the additional inverse bias correction, though these adjustments offer only minor improvements compared to higher momentum.

# 7 The Detrimental Effects of Large Updates

In appx. A we empirically investigate potential causes for the lasting performance degradation from large initial updates for a small ResNet model. We find that the performance impact best correlates with the number of dead ReLUs and is improved by the use of leaky-ReLUs, which fits well with our perspective of large changes in the internal representations. We also investigated whether overfitting to initial training samples or the correlation between weight vectors of different neurons could explain the necessity for warmup, but these factors did not show a significant impact.

# 8 The Role of Larger Batch Sizes

Warmup is often used with larger batch sizes in particular, for example in the setting where Goyal et al. [10] first proposed using linear warmup. Although this was for SGD, we expect the need for warmup to be amplified at larger batch sizes for two of the reasons we identified. The first is that larger batch sizes are more likely to exceed the critical batch size early in training. The second is the size of early angular updates. As shown by Kosson et al. [19], the equilibrium weight magnitude depends on the learning rate and weight decay value. Common hyperparameter scaling rules for a modified batch size only change the learning rate but not the weight decay, which shifts the equilibrium magnitude. The smaller the initialization magnitude is compared to the equilibrium magnitude, the larger the early angular updates will be relative to their steady state value, potentially necessitating warmup.

# 9 Limitations

Our main experiments focus on a single network which may not be broad enough to generalize to a wide range of networks. In appx. C.2 we experiments with an additional dataset and architecture but the scale of the experiments is still limited and they cover a limited range of hyperparameters. We believe we identify real factors that contribute to the need for warmup, but these may not be the only

ones across a broader range of settings. Similarly, the promising results for reducing or eliminating the warmup with higher momentum values or the relative representation correction would benefit from broader validation.

## 10 Conclusion

In this work we explored how the size of the updates $\Delta w = \eta u$ impacts the need for learning rate warmup. We showed that $u$ can be large initially when measured in terms of its $\ell_2$-norm (§4), the resulting directional change in $w$ (angular update, §5), as well as the resulting change in the internal representations of the network (relative representation change, §6). We argued that small initial values of the learning rate $\eta$ are beneficial to counteract large values of $u$, i.e. that a learning rate warmup simply keeps some notion of the overall "effective" update size reasonable. We showed this experimentally rather than theoretically by modifying the optimizers to normalize the size of $u$ based on each metric and measuring how these changes affected the benefit of using learning rate warmup.

The two weight-based measures of the update size, the $\ell_2$-norm and angular update did not fully account for the need for warmup. However, quantifying the update size in terms of the relative change in neural representations shows potential. This measure is closely linked to the angular update size but accounts for changes in the signal characteristics of the gradient, which can vary significantly throughout training. Effectively controlling neural representation changes is a challenging task we leave for future work, but our initial attempts show encouraging results in reducing the need for a manually configured warmup. We also highlighted the importance of high momentum for warmup; when combined with angular update control and an inverse bias correction, it may enable efficient warmup-free training. Overall, our work provides new insights into the benefit of learning rate warmup with modern optimizers beyond SGD and suggests potential directions for eliminating it.

Although we present new methods we consider promising, we still recommend the use of a short warmup in practice. Fully eliminating it seems to require significant modifications that also need further validation across additional settings. However, we hope to have provided the reader with a new perspective and simple intuition for why warmup is beneficial for training. We also hope our work inspires further exploration of how learning should be controlled and scheduled in neural network training. In particular, it seems that the learning rate in current optimizers does not really control the "rate of learning", making learning rate schedules and the use of warmup highly arbitrary.

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

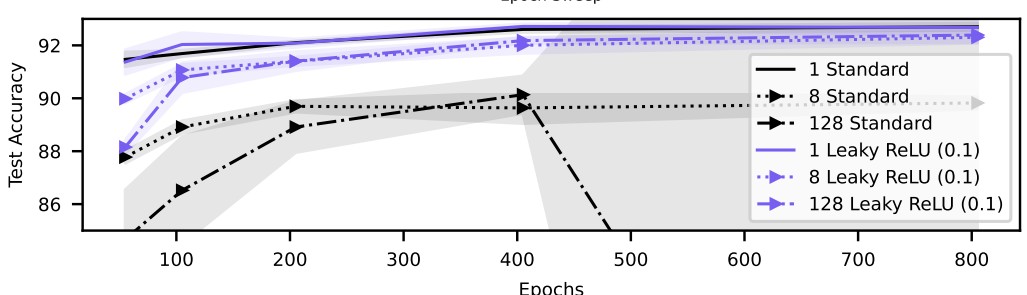

**Figure 6:** The performance gap caused by large initial updates persists despite extended training (800 epochs) in a standard ResNet-20. We investigate the influence of network non-linearities by comparing two training methods while scaling update sizes during the 5 epoch initial phase by factors of 1, 8, and 128: *Standard (S)*, which employs traditional ReLU activations, and *Leaky ReLU*, which replaces ReLUs with Leaky ReLUs using a scaling factor of $\alpha = 0.1$. We observe that training with *Leaky ReLU* results in smaller performance degradation from large initial updates, suggesting that the non-linearities in the network might substantially impact the observed performance degradation.

## A    The Detrimental Effects of Large Updates

To investigate the effects of large updates at the beginning of training, we conducted controlled experiments on a ResNet-20 model on CIFAR-10 [20] due to resource constraints. We controlled the average angular update throughout training using the rotational optimizer variant of AdamW proposed by Kosson et al. [19]. For the initial phase of training, 5 epochs, we either use a standard learning rate of 0.05 or amplify it by a factor of 8 or 128. This results in very large updates, scaling the rotation by either $\sqrt{8}$ or $\sqrt{128}$. For all experiments, we used a weight decay of 0.01, $\beta_1 = 0.9$, $\beta_2 = 0.999$, 5 epoch initial phase, and trained for 205 epochs in total with a cosine schedule unless otherwise specified. The data was pre-processed by normalizing it with a mean of $(0.4914, 0.4822, 0.4465)$ and a standard deviation of $(0.2023, 0.1994, 0.2010)$ and applying simple data augmentation techniques as described by He et al. [11]. To run the experiment, we used the codebase from Wightman [40] and extended the utilities from Kosson et al. [19].

As shown in fig. 6, the performance of standard training does not recover when large updates are used at the beginning of training, even when the training time is extended to four times the normal duration for ReLU networks. This suggest that large, controlled, initial updates can result in a permanent performance degradation, similar to what we observe without warmup in advanced settings. The impact is much smaller when replacing ReLUs with leaky ReLUs, suggesting that the non-linearities in the network might substantially contribute to the performance degradation.

In fig. 7 we measure the fraction of dead ReLUs directly across different settings and scaling factors. We find that large initial updates do indeed lead to a large number of permanently dead units and that the final accuracy suffers when this is the case. This effect can be mitigated by freezing the biases at the beginning of training, as shown in the table in fig. 7. We also observe that replacing the actual gradients with random gradients has a much smaller impact, suggesting that the direction of the updates also matters for the degradation, not only their size.

Interestingly, we did not find a connection to overfitting to a small number of samples at the beginning of training. The performance of 92.1 can be recovered in this case. Additionally, we explored stable rank measurements as a potential factor but did not find a significant relation, as detailed in fig. 8.

## B    Additional Mathematical and Technical Details

### B.1    The magnitude of the Momentum Vector

Let's assume a scalar gradient $g_t$ (e.g. for some coordinate) that is a random variable that is independent across time and has a zero mean distribution that does not change across time, i.e. $E[g_t] = 0$

| Method | Scale 1 | Scale 8 | Scale 128 |
|---|---|---|---|
| Standard | 92.13±0.2 | 89.48±0.2 | 88.22±1.6 |
| Standard frozen bias | 92.30±0.3 | 92.08±0.3 | 92.30±0.2 |
| Random | 92.05±0.3 | 91.74±0.2 | 89.54±0.3 |
| Random frozen bias | 92.27±0.2 | 92.12±0.4 | 92.20±0.1 |
| Leaky ReLU | 92.16±0.3 | 91.48±0.3 | 91.82±0.4 |
| Leaky ReLU frozen bias | 92.46±0.2 | 92.49±0.1 | 92.35±0.2 |

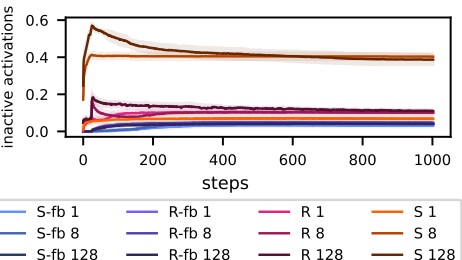

**Figure 7:** Comparison of the performance (final test accuracy) and fraction of dead ReLUs (inactive activations) across different settings. The learning rate in the initial phase of 5 epochs is scaled by a factor of either 1, 8, or 128. *Standard (S)* denotes normal training, while *Frozen Biases (fb)* involves freezing the biases at the onset of training. The *Random (R)* approach employs random gradient directions at the start of training, and *Leaky ReLU* replaces the ReLUs in standard models with Leaky ReLUs using a scaling factor of $\alpha = 0.1$. We observe a notable correspondence between large initial updates, higher ratios of dead ReLUs in ResNet-20, and performance degradation.

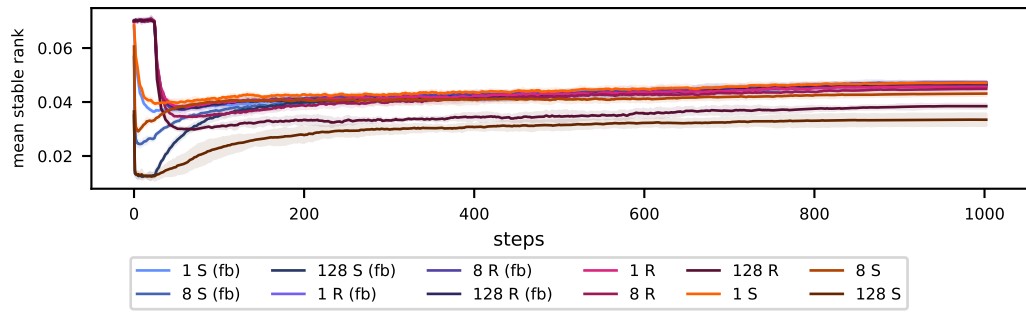

**Figure 8:** Impact of varying update sizes during the warmup phase on the stable rank of a standard ResNet-20. The learning rate in the initial phase of 5 epochs is scaled by a factor of either 1, 8, or 128. We evaluate the effects of across different training configurations: *Standard (S)* denotes normal training; *Frozen Biases (fb)* involves freezing the biases at the onset of training; and *Random (R)* employs random gradient directions at the start of training. The stable rank remains largely consistent across these methods, except when using extremely large updates—specifically, scaling the learning rate by a factor of 128 without freezing biases—which leads to noticeable variations in the rank.

and $\mathbb{E}[g_t^2] = \sigma^2$. For standard **heavyball-momentum** $m_t$, with $m_0 = 0$ and coefficient $\beta$ (equivalent to $\beta_1$ for Adam) we have:

$$\mathbb{E}[m_t^2] = \mathbb{E}[(\beta m_{t-1} + (1-\beta)g_t)^2] \tag{10}$$

$$= \mathbb{E}\left[\left((1-\beta)\sum_{i=0}^{t-1}\beta^i g_{t-i}\right)^2\right] \tag{11}$$

$$= \mathbb{E}\left[(1-\beta)^2\sum_{i=0}^{t-1}\beta^{2i}g_{t-i}^2 + (1-\beta)^2\sum_{j=0}^{t-1}\sum_{\substack{k=0\\k\neq j}}^{t-1}\beta^{2t-j-k}g_{t-j}g_{g-k}\right] \tag{12}$$

$$= (1-\beta)^2\sum_{i=0}^{t-1}\beta^{2i}\mathbb{E}[g_{t-i}^2] + (1-\beta)^2\sum_{j=0}^{t-1}\sum_{\substack{k=0\\k\neq j}}^{t-1}\beta^{2t-j-k}\mathbb{E}[g_{t-j}]\mathbb{E}[g_{g-k}] \tag{13}$$

$$= (1-\beta)^2\sum_{i=0}^{t-1}\beta^{2i}\sigma^2 + 0 \tag{14}$$

$$= (1 - \beta)^2 \frac{1 - \beta^{2t}}{1 - \beta^2} \sigma^2 \tag{15}$$

$$= (1 - \beta)^2 \frac{1 - \beta^{2t}}{(1 - \beta)(1 + \beta)} \sigma^2 \tag{16}$$

$$= (1 - \beta^{2t}) \frac{1 - \beta}{1 + \beta} \sigma^2 \tag{17}$$

In the limit $t \to \infty$ we have $(1 - \beta^{2t}) \to 1$. We can derive the size of the second-moment $v_t$ in AdamW in an analogous way, obtaining $\mathbb{E}[v_t] = (1 - \beta_2^t)\sigma^2$. For a random walk, the update size of Adam is scaled in a similar way. Since the update size of Lion is fixed and does not depend on $\beta$, we scale the update size to match that of AdamW for a random walk in a steady state, i.e. by $\gamma = \sqrt{\frac{1-\beta}{1+\beta}}$ as seen in algo. 2.

For **Nesterov momentum**, the update is modified to use:

$$u_t = \beta m_t + (1 - \beta)g_t \tag{18}$$

$$= \beta \left( \beta m_{t-1} + (1 - \beta)g_t \right) + (1 - \beta)g_t \tag{19}$$

$$= \beta^2 m_{t-1} + (1 - \beta)(1 + \beta)g_t \tag{20}$$

Note that $m_{t-1}$ and $g_t$ are independent and zero-mean, allowing us to use the previous result for:

$$\mathbb{E}[u_t^2] = \mathbb{E}\left[ \left( \beta^2 m_{t-1} + (1 - \beta)(1 + \beta)g_t \right)^2 \right] \tag{21}$$

$$= \beta^4 \mathbb{E}[m_{t-1}^2] + (1 - \beta^2)^2 \mathbb{E}[g_t^2] \tag{22}$$

$$= \beta^4 (1 - \beta^{2t-2}) \frac{1 - \beta}{1 + \beta} \sigma^2 + (1 - \beta^2)^2 \sigma^2 \tag{23}$$

In the limit $t \to \infty$ this gives the Nesterov scaling factor used in LionA (algo. 2) to ensure that the update size corresponds to that of AdamW using an analogous Nesterov update.

**Inverse bias correction for momentum.** Adam uses a bias correction to attempt to fix the update size over time. This scales early updates resulting in the contributions of the corresponding gradients being amplified. The relative representation change for those samples is increased as a result, similar to applying the same update multiple times. Removing the $\beta_1$ bias correction from AdamW removes this effect. LionA and LionAR similarly scale the update size, making it constant. We can counteract this by changing our scaling factors to use the time varying expressions based on the derivations above. Note however, that this assumed the gradients were uncorrelated so it only approximately undoes the scaling effect for real values with arbitrary alignment of successive gradients. To summarize, the inverse bias correction for momentum changes the momentum scaling factors ($\gamma$ in algo. 3) to vary over time:

$$\text{Nesterov:} \quad \gamma_t = \sqrt{(1 - \beta^2)^2 + (1 - \beta^{2t-2})\beta^4 \frac{1 - \beta}{1 + \beta}} \tag{24}$$

$$\text{Heavy-ball:} \quad \gamma_t = \sqrt{(1 - \beta^{2t}) \frac{1 - \beta}{1 + \beta}} \tag{25}$$

## B.2 Properties of Scale Invariance

Derivations for the gradient magnitude and curvature can be found in existing works, for example Lyu et al. [25]. When a scale invariant weight is scaled by a factor $c > 0$, the gradient is scaled by $c^{-1}$ which scales the ratio of the gradient norm and weight norm, and therefore the angular updates, by $c^{-2}$. For normalized optimizers like Adam and Lion, where the update norm is not affected by the gradient magnitude, this factor is decreased to $c^{-1}$.

## B.3 The Angular Update Size in LionAR

The scaling factor for the angular update size in algo. 3 is adopted directly from the AdamW value derived by Kosson et al. [19]. Since the Nesterov momentum does not change the total contribution of each gradient it does not affect the equilibrium magnitude. The expected angular updates are therefore scaled in the same way as the RMS update norm we derived in appx. B.1.

## B.4 Relative Representation Change for Normalized Gradient Descent

**Property (P1):** For a dot product $y = \langle \boldsymbol{w}, \boldsymbol{x} \rangle$ and loss $\mathscr{L}(\boldsymbol{x}_b)$ that depends on $y$, we have:

$$\frac{\partial \mathscr{L}(\boldsymbol{x}_b)}{\partial \boldsymbol{w}} = \frac{\partial \mathscr{L}(\boldsymbol{x}_b)}{\partial y} \frac{\partial y}{\partial \boldsymbol{w}} = \frac{\partial \mathscr{L}(\boldsymbol{x}_b)}{\partial y} \boldsymbol{x}_b \tag{26}$$

where $\frac{\partial \mathscr{L}(\boldsymbol{x}_b)}{\partial y}$ is a scalar, ensuring that $\boldsymbol{g}_b := \frac{\partial \mathscr{L}(\boldsymbol{x}_b)}{\partial \boldsymbol{w}} \parallel \boldsymbol{x}_b$, assuming the vectors are not zero.

**Lemma (L1):** Consider two independent random vectors $\boldsymbol{a} \in \mathbb{R}^C$ and $\boldsymbol{b} \in \mathbb{R}^C$, whose elements are independent and identically distributed (IID). If at least one of the vectors has a zero-mean distribution, then the expected value of the squared inner product of $\boldsymbol{a}$ and $\boldsymbol{b}$ is given by:

$$\mathbb{E}[\langle \boldsymbol{a}, \boldsymbol{b} \rangle^2] = \frac{\mathbb{E}[\|\boldsymbol{a}\|^2]\mathbb{E}[\|\boldsymbol{b}\|^2]}{C} \tag{27}$$

**Proof**: Let $\boldsymbol{a} = (a_1, a_2, \ldots, a_C)$ and $\boldsymbol{b} = (b_1, b_2, \ldots, b_C)$. The inner product $\langle \boldsymbol{a}, \boldsymbol{b} \rangle$ is given by:

$$\langle \boldsymbol{a}, \boldsymbol{b} \rangle = \sum_{i=1}^{C} a_i b_i.$$

We need to find $\mathbb{E}[\langle \boldsymbol{a}, \boldsymbol{b} \rangle^2]$. Expanding the square of the inner product:

$$\langle \boldsymbol{a}, \boldsymbol{b} \rangle^2 = \left( \sum_{i=1}^{C} a_i b_i \right)^2 = \sum_{i=1}^{C} \sum_{j=1}^{C} a_i b_i a_j b_j.$$

Taking the expectation, we get:

$$\mathbb{E}[\langle \boldsymbol{a}, \boldsymbol{b} \rangle^2] = \mathbb{E}\left[ \sum_{i=1}^{C} \sum_{j=1}^{C} a_i b_i a_j b_j \right] = \sum_{i=1}^{C} \sum_{j=1}^{C} \mathbb{E}[a_i b_i a_j b_j].$$

Since $\boldsymbol{a}$ and $\boldsymbol{b}$ are independent and their elements are IID, we have:

$$\mathbb{E}[a_i b_i a_j b_j] = \mathbb{E}[a_i a_j]\mathbb{E}[b_i b_j].$$

Consider two cases:

1. When $i = j$:
$$\mathbb{E}[a_i b_i a_i b_i] = \mathbb{E}[a_i^2]\mathbb{E}[b_i^2].$$

2. When $i \neq j$:
$$\mathbb{E}[a_i b_i a_j b_j] = \mathbb{E}[a_i]\mathbb{E}[b_i]\mathbb{E}[a_j]\mathbb{E}[b_j].$$

Given that at least one of $\boldsymbol{a}$ or $\boldsymbol{b}$ has a zero-mean distribution, say $\boldsymbol{a}$ without loss of generality, we have $\mathbb{E}[a_i] = 0$. Thus:
$$\mathbb{E}[a_i b_i a_j b_j] = 0.$$

So, the expectation simplifies to:

$$\mathbb{E}[\langle \boldsymbol{a}, \boldsymbol{b} \rangle^2] = \sum_{i=1}^{C} \mathbb{E}[a_i^2]\mathbb{E}[b_i^2].$$

Since $a_i$ and $b_i$ are IID, we have:

$$\mathbb{E}[a_i^2] = \mathbb{E}[a_1^2] \quad \text{and} \quad \mathbb{E}[b_i^2] = \mathbb{E}[b_1^2].$$

Therefore:

$$\mathbb{E}[\langle \boldsymbol{a}, \boldsymbol{b} \rangle^2] = C\mathbb{E}[a_1^2]\mathbb{E}[b_1^2].$$

Recognizing that:

$$\mathbb{E}[\|\boldsymbol{a}\|^2] = \mathbb{E}\left[\sum_{i=1}^{C} a_i^2\right] = C\mathbb{E}[a_1^2],$$

$$\mathbb{E}[\|\boldsymbol{b}\|^2] = \mathbb{E}\left[\sum_{i=1}^{C} b_i^2\right] = C\mathbb{E}[b_1^2],$$

we have:

$$\mathbb{E}[a_1^2] = \frac{\mathbb{E}[\|\boldsymbol{a}\|^2]}{C} \quad \text{and} \quad \mathbb{E}[b_1^2] = \frac{\mathbb{E}[\|\boldsymbol{b}\|^2]}{C}.$$

Thus:

$$\mathbb{E}[\langle \boldsymbol{a}, \boldsymbol{b}\rangle^2] = C\left(\frac{\mathbb{E}[\|\boldsymbol{a}\|^2]}{C}\right)\left(\frac{\mathbb{E}[\|\boldsymbol{b}\|^2]}{C}\right) = \frac{\mathbb{E}[\|\boldsymbol{a}\|^2]\mathbb{E}[\|\boldsymbol{b}\|^2]}{C}.$$

This completes the proof.

**Assumption (A1):** We assume the following vector pairs satisfy L1: $(\boldsymbol{x}_i, \tilde{\boldsymbol{g}}_b)$ when $i \neq b$, $(\bar{\boldsymbol{g}}, \tilde{\boldsymbol{g}}_b)$ and $(\boldsymbol{w}, \boldsymbol{x}_b)$.

Vector pairs of the type $(\boldsymbol{x}_i, \tilde{\boldsymbol{g}}_b)$ and $(\bar{\boldsymbol{g}}, \tilde{\boldsymbol{g}}_b)$ should be independent and $\tilde{\boldsymbol{g}}_b$ has a zero mean distribution. However, the elements of each vector are not necessarily IID. For $(\boldsymbol{w}, \boldsymbol{x}_b)$, this is an even stronger assumption. Generally, neither $\boldsymbol{w}$ nor $\boldsymbol{x}_b$ is guaranteed to be IID or zero mean, and their independence later in training does not necessarily hold. Applying weight standardization to $\boldsymbol{w}$ or batch normalization to $\boldsymbol{x}$ would suffice to make their mean zero. Overall, this assumption can be viewed as a simplifying approximation to obtain reasonable predictions without additional information about these vectors. Everett et al. [5] explore the behavior of $\langle \boldsymbol{w}, \boldsymbol{x}_b\rangle$ throughout training and find that it is more complicated than assumed here. This will lead to additional factors that may affect the RRC but we do not attempt to analyze.

**Deriving the Relative Representation Change:** Applying L1 directly gives us the original expected square output :

$$\mathbb{E}[y_b^2] = \mathbb{E}[\langle \boldsymbol{w}, \boldsymbol{x}_b\rangle^2] = \frac{\|\boldsymbol{w}\|^2\mathbb{E}[\|\boldsymbol{x}_b\|^2]}{C} \tag{28}$$

For the expected square representation change we get:

$$\mathbb{E}[(\Delta y_b)^2] \tag{29}$$

$$= \mathbb{E}[\langle -\eta\boldsymbol{g}/\sqrt{\mathbb{E}[\|\boldsymbol{g}\|^2]}, \boldsymbol{x}_b\rangle^2] \tag{30}$$

$$= \frac{\eta^2}{B^2}\frac{1}{\mathbb{E}[\|\boldsymbol{g}\|^2]}\mathbb{E}\left[\left(\sum_{i=1}^{B}\langle \boldsymbol{g}_i, \boldsymbol{x}_b\rangle\right)^2\right] \tag{31}$$

$$= \frac{\eta^2}{B^2}\frac{1}{\mathbb{E}[\|\boldsymbol{g}\|^2]}\mathbb{E}\left[\left(\text{sign}(\langle \boldsymbol{x}_b, \boldsymbol{g}_b\rangle)\|\boldsymbol{g}_b\|\|\boldsymbol{x}_b\| + \sum_{i\neq B}\langle \boldsymbol{g}_i, \boldsymbol{x}_b\rangle\right)^2\right] \tag{32}$$

$$= \frac{\eta^2}{B^2}\frac{1}{\mathbb{E}[\|\boldsymbol{g}\|^2]}\mathbb{E}\left[\left(\text{sign}(\langle \boldsymbol{x}_b, \boldsymbol{g}_b\rangle)\|\boldsymbol{g}_b\|\|\boldsymbol{x}_b\| + (B-1)\langle \bar{\boldsymbol{g}}, \boldsymbol{x}_b\rangle + \sum_{i\neq b}\langle \tilde{\boldsymbol{g}}_i, \boldsymbol{x}_b\rangle\right)^2\right] \tag{33}$$

$$\tag{34}$$

where we have used the definitions from eq. (7) and D1. Using property P1, we can write:

$$\langle \bar{\boldsymbol{g}}, \boldsymbol{x}_b\rangle = \left\langle \bar{\boldsymbol{g}}, \quad \text{sign}(\langle \boldsymbol{x}_b, \boldsymbol{g}_b\rangle)\frac{\|\boldsymbol{x}_b\|}{\|\boldsymbol{g}_b\|} \cdot (\bar{\boldsymbol{g}} + \tilde{\boldsymbol{g}}_b)\right\rangle \tag{35}$$

$$= \text{sign}(\langle \boldsymbol{x}_b, \boldsymbol{g}_b\rangle)\frac{\|\boldsymbol{x}_b\|}{\|\boldsymbol{g}_b\|}(\|\bar{\boldsymbol{g}}\|^2 + \langle \bar{\boldsymbol{g}}, \tilde{\boldsymbol{g}}_b\rangle) \tag{36}$$

Plugging this into the previous expression yields $\mathbb{E}[(\Delta y_b)^2]$

$$= \frac{\eta^2}{B^2} \frac{1}{\mathbb{E}[\|\boldsymbol{g}\|^2]} \mathbb{E}\left[\left(\text{sign}(\langle \boldsymbol{x}_b, \boldsymbol{g}_b\rangle)\left(\|\boldsymbol{g}_b\|\|\boldsymbol{x}_b\| + (B-1)\frac{\|\boldsymbol{x}_b\|}{\|\boldsymbol{g}_b\|}(\|\bar{\boldsymbol{g}}\|^2 + \langle \bar{\boldsymbol{g}}, \tilde{\boldsymbol{g}}_b\rangle)\right) + \sum_{i\neq b}\langle \tilde{\boldsymbol{g}}_i, \boldsymbol{x}_b\rangle\right)^2\right]$$

(37)

Squaring the expression results in various cross but all remaining dot products except the sign one are zero in expectation (due to the noise $\tilde{g}$) and independent from each other. The cross terms involving these thus all disappear under the expectation. We apply Lemma L1 to their squares and approximate the expected norms of $\boldsymbol{x}_b$ and $\boldsymbol{g}_b$ as being independent. This gives $\mathbb{E}[(\Delta y_b)^2]$

$$= \frac{\eta^2}{B^2}\frac{\mathbb{E}[\|\boldsymbol{x}_b\|^2]}{\mathbb{E}[\|\boldsymbol{g}\|^2]}\left(\mathbb{E}[\|\boldsymbol{g}_b\|^2] + \frac{(B-1)^2\|\bar{\boldsymbol{g}}\|^2}{\mathbb{E}[\|\boldsymbol{g}\|^2]}\left(\|\bar{\boldsymbol{g}}\|^2 + \frac{\mathbb{E}[\|\tilde{\boldsymbol{g}}_b\|^2]}{C}\right)\right.$$

(38)

$$\left. +2(B-1)\|\bar{\boldsymbol{g}}\|^2 + \frac{B-1}{C}\mathbb{E}[\|\tilde{\boldsymbol{g}}_i\|^2]\right)$$

(39)

We can compute the expected magnitude of the batch gradient as:

$$\mathbb{E}[\|\boldsymbol{g}\|^2] = \mathbb{E}[\|\frac{1}{B}\sum_{i=1}^{B}(\bar{\boldsymbol{g}}+\tilde{\boldsymbol{g}}_i)\|^2] = \mathbb{E}[\|(\bar{\boldsymbol{g}}+\frac{1}{B}\sum_{i=1}^{B}\tilde{\boldsymbol{g}}_i)\|^2] = \|\bar{\boldsymbol{g}}\|^2 + \frac{1}{B}\mathbb{E}[\|\boldsymbol{g}_i\|^2]$$

(40)

and similarly $\mathbb{E}[\|\boldsymbol{g}_b\|^2] = \|\bar{\boldsymbol{g}}\|^2 + \mathbb{E}[\|\tilde{\boldsymbol{g}}_b\|^2]$. Using these facts we can further write $\mathbb{E}[(\Delta y_b)^2]$

$$= \frac{\eta^2}{B^2}\frac{\mathbb{E}[\|\boldsymbol{x}_b\|^2]}{\mathbb{E}[\|\bar{\boldsymbol{g}}\|^2] + \frac{1}{B}\mathbb{E}[\|\boldsymbol{g}_i\|^2]}\left(\|\bar{\boldsymbol{g}}\|^2 + \mathbb{E}[\|\tilde{\boldsymbol{g}}_b\|^2] + \frac{(B-1)^2\|\bar{\boldsymbol{g}}\|^2}{\|\bar{\boldsymbol{g}}\|^2 + \mathbb{E}[\|\tilde{\boldsymbol{g}}_b\|^2]}\left(\|\bar{\boldsymbol{g}}\|^2 + \frac{\mathbb{E}[\|\tilde{\boldsymbol{g}}_b\|^2]}{C}\right)\right.$$

$$\left. +2(B-1)\|\bar{\boldsymbol{g}}\|^2 + \frac{B-1}{C}\mathbb{E}[\|\tilde{\boldsymbol{g}}_i\|^2]\right)$$

(41)

Combining this with the previous expression for $\mathbb{E}[y_b^2]$ and the definition (D2) of the signal-to-noise ratio $\varphi := \mathbb{E}[\|\bar{\boldsymbol{g}}\|^2]/\mathbb{E}[\|\tilde{\boldsymbol{g}}_b\|^2]$ we obtain the expression in the main body:

$$\frac{\mathbb{E}[(\Delta y_b)^2]}{\mathbb{E}[y_b^2]} = \frac{\eta^2 C}{B^2\|\boldsymbol{w}\|^2}\frac{1}{\mathbb{E}[\|\boldsymbol{g}\|^2]}\left(\mathbb{E}[\|\boldsymbol{g}_b\|^2] + \frac{B-1}{C}\mathbb{E}[\|\tilde{\boldsymbol{g}}_i\|^2]\right.$$

$$\left. + \frac{(B-1)^2}{\mathbb{E}[\|\boldsymbol{g}_b\|^2]}\left(\|\bar{\boldsymbol{g}}\|^4 + \frac{\|\bar{\boldsymbol{g}}\|^2\mathbb{E}[\|\tilde{\boldsymbol{g}}_b\|^2]}{C}\right) + 2(B-1)\|\bar{\boldsymbol{g}}\|^2\right)$$

(42)

$$= \frac{\eta^2 C}{B^2\|\boldsymbol{w}\|^2}\frac{1}{\varphi+\frac{1}{B}}\left((\varphi+1) + \frac{B-1}{C} + \left(\frac{(B-1)^2\varphi}{\varphi+1}\left(\varphi+\frac{1}{C}\right) + 2(B-1)\varphi\right)\right)$$

(43)

### B.5    Estimating the Signal-to-Noise Ratio

We use accumulation over the microbatches to estimate the SNR at a given time. Let's assume we have $A$ microbatches of size $M$ each, with the average gradient of a microbatch denoted $\boldsymbol{g}_m$ and the average gradient of the whole batch denoted $\boldsymbol{g} = \frac{1}{A}\sum_m \boldsymbol{g}_m$.

We estimate the variance of the norm of a single gradient example, i.e. the noise power as:

$$P_N = \frac{A}{A-1}\cdot M\cdot \mathbf{1}^{\top}\left(\frac{1}{A}\sum_m \boldsymbol{g}_m^2 - \boldsymbol{g}^2\right)$$

(44)

The signal power is estimated as:

$$P_S = \mathbf{1}^{\top}\boldsymbol{g}^2 - \frac{1}{AM}P_N$$

(45)

Our SNR estimate is then:

$$\varphi = P_S/P_N$$

(46)

## B.6 RRC Correction Factor

The RRC correction is done based on eq. (9) and the SNR estimation eq. (46). We assume the learning rate was originally scaled with the square root of the batch size, which is derived for an SNR of zero, and downscale the step size to compensate for the measured SNR and batch size. We define:

$$\rho = \frac{1}{B(1+\varphi)}\left((\varphi+1) + \frac{B-1}{C} + \left(\frac{(B-1)^2\varphi}{\varphi+1}\left(\varphi + \frac{1}{C}\right) + 2(B-1)\varphi\right)\right) \qquad (47)$$

For numerical purposes, we clamp $1 \leq \rho \leq B$ which corresponds to $\varphi = 0$ and $\varphi = \infty$ for a large $C \to \infty$. The update scaling factor is the square root of an EMA of the inverse of this quantity. We use the same coefficient as for the momentum and compute this for the matrix of each linear layer independently. This form for the scaling factor is somewhat arbitrary, complicated by the fact that Lion-like algorithms fix the step size exactly, so scaling the gradient at each step size can not change the magnitude of the update. For Adam or SGD like algorithms we could scale the gradient contributions directly instead of scaling the update size.

## B.7 Run-to-run Variance / Uncertainty Estimation

We do not quantify the uncertainty for every GPT2 configuration in our sweeps. This would require significantly more compute and our estimates of the uncertainty for select points indicate that this would not qualitatively change our results. For the baseline AdamW run the run-to-run differences in the validation loss over different seeds are around 0.05. However, the relative ranking of different runs remained the same.

## B.8 Computational Requirements

Our experiments are performed on A100 GPUs with either 40GB or 80GB of RAM. One training run for our GPT2 setup takes around 4h, running on a single GPU. Reproducing the GPT2 experiments reported in the main body should take on the order of 1000 GPU hours. Including our preliminary experiments brings this up to around 3x this amount.

# C   Additional Experiments

## C.1   Comparison with RAdam

RAdamW combines the variance reduction technique of Liu et al. [23] with the decoupled weight decay of Loshchilov and Hutter [24]. Since it is a well known technique for reducing the need for warmup it serves as a good comparison and contextualization of our work. Figure 9 shows that while RAdamW outperforms the AdamW baseline without warmup, it is unable to match longer warmups. Ma and Yarats [26] suggests that RAdamW is approximately equivalent to 4 steps of SGDM followed by AdamW with a special type of built-in warmup with an effective length of around $2/(1 - \beta_2) = 40$, which is likely too short in our setting. For comparison the 2% warmup shown corresponds to 100 steps but is too short for optimal performance.

The analysis of Liu et al. [23] is based on the idea that early in training the second-moment estimates are not accurate (noisy) and can therefore not be trusted to scale the update properly. This could in turn contribute to the need for warmup, although Ma and Yarats [26] question this interpretation. We first note that without momentum, perfect estimates of the second moment at the current time step would control the expected $\ell_2$-norm of the update. This relates our approach of looking at the update size to the adaptive learning rate view of Liu et al. [23]. Secondly, we note that counteracting noisy estimates of the second moment can not be the sole reason warmup is beneficial. This is supported by the fact that both SGD and Lion empirically need warmup in various settings but do not use the second moment at all, indicating there are additional factors that contribute to the need for warmup.

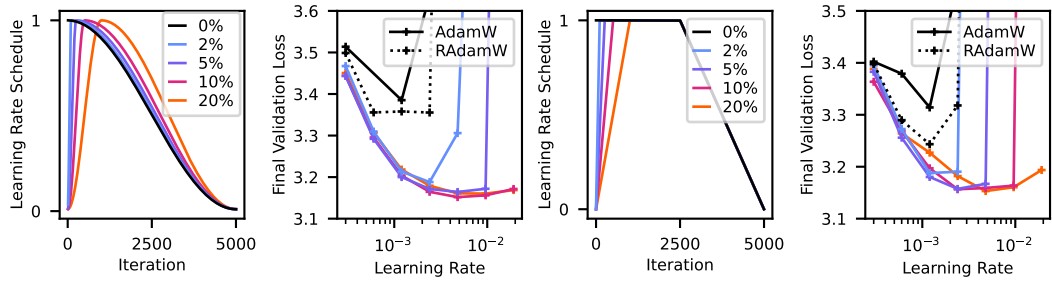

**Figure 9:** Comparison of the AdamW baseline with a cosine schedule and RAdamW. **Panel 1:** Cosine schedules with different warmup lengths. Note how the warmup shifts the curve, affecting the whole schedule including the decay portion. **Panel 2:** The final validation loss for GPT2-124M training on OpenWebText using the cosine schedules. Note that RAdamW helps but does not eliminate the need for warmup. **Panel 3:** The original trapezoidal schedules used in our experiments. **Panel 4:** Trapezoidal GPT2-124M OpenWebText results. The RAdamW results are similar to those in panel 2.

## C.2   Model & Dataset Ablations

In this section we repeat some of our main experiments, varying the dataset and model architecture.

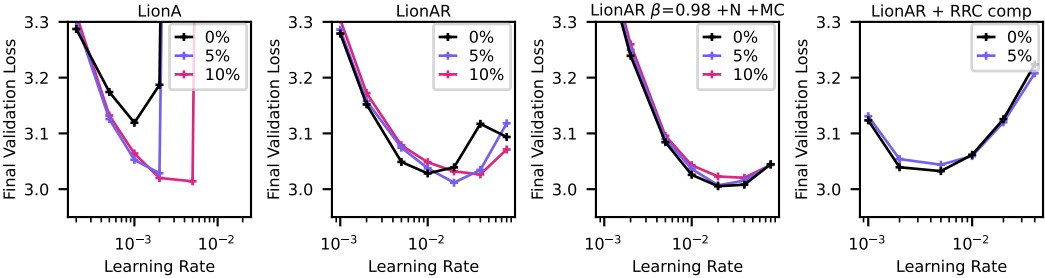

**Figure 10:** Dataset ablation study, GPT2-124M on SlimPajama [33]. Four update control approaches are shown (see titles). The overall results are similar to those for OpenWebText in the manuscript.

Figure 10 shows the effects of changing the dataset used in our experiments from OpenWebText [8] to SlimPajama [33]. Overall the results are similar as before, when controlling the $\ell_2$-norm via LionA warmup is still beneficial, controlling the angular updates via LionAR decreases the gap significantly. The higher momentum LionAR with Nesterov momentum and our momentum correction eliminates the gap fully. The RRC also seems to eliminate the benefit of warmup but still has the same practical limitations as we describe in §6.

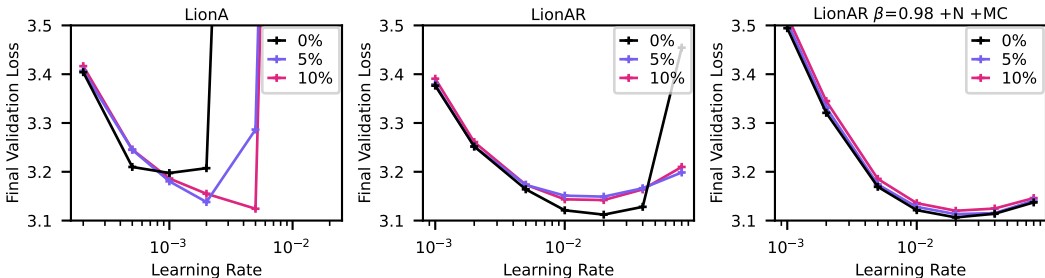

**Figure 11:** Architecture ablation, Llama2-124M on OpenWebText. Here LionAR is already sufficient to eliminate the need for warmup, no additional RRC compensation is needed. This could be due to the critical batch size being larger (for unclear reasons).

Figure 11 shows the effects of changing the architecture from GPT2 to a Llama style [36] while keeping the dataset and parameter count (124m) the same. This change consists of using SwiGLU activations, RoPE embeddings and RMSNorm. In this case LionAR is able to fully eliminate the need for warmup without any additional tricks like the RRC compensation or momentum corrections. Based on our analysis these additional tricks are likely only needed when the critical batch size is very small initially. We expect that using larger batch sizes could necessitate these additional tricks for Llama, but do not explore this further here.

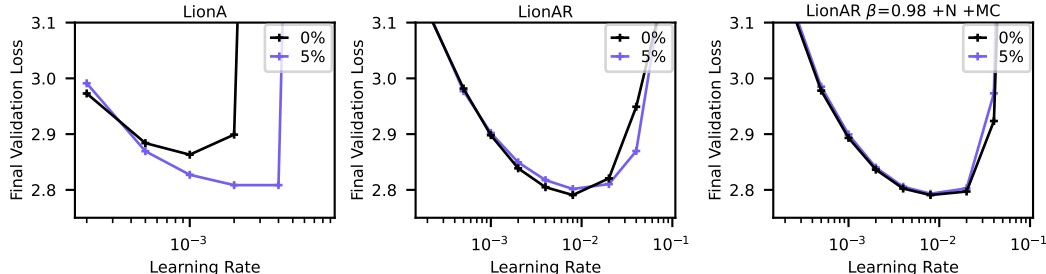

**Figure 12:** Larger llama2-209M training on SlimPajama. This experiment changes the architecture, dataset, model size, and training length (proportional to the model size). In this case LionAR suffices on its own again, no additional RRC correction needed as in the smaller llama experiments.

Figure 12 uses a larger Llama model with twice the depth. It also trains for twice as many iterations to keep the ratio of tokens to parameters similar. The overall results resemble those of the smaller llama experiments in fig. 11.

