# OpenReview forum: "Analyzing & Reducing the Need for Learning Rate Warmup in GPT Training"
_NeurIPS.cc/2024/Conference — NeurIPS 2024 poster_

### Official Review · Reviewer_ramW · 2024-07-11

**Soundness:** 3
**Presentation:** 2
**Contribution:** 3
**Rating:** 6
**Confidence:** 3

**Summary:**

This work explores the benefits of learning rate warmup in neural network training, focusing on the size of model updates via the GPT2 model. It finds that controlling update size in parameter space doesn't fully explain warmup's advantages, but quantifying updates in terms of neural representation changes shows promise. The study also highlights the role of high momentum in warmup and suggests potential methods for reducing the need for manual warmup configuration. Overall, the research provides insights into learning rate warmup's necessity and potential ways to eliminate it in practice.

**Strengths:**

The paper addresses an intriguing topic, which aims to present a systematic understanding regarding the LR warmup heuristic from a novel perspective.  However, I feel that the authors have attempted to cover too many aspects, which might be challenging to thoroughly demonstrate within the scope of a single conference paper.

**Weaknesses:**

1. I noticed that the authors have not adequately discussed the highly relevant paper, "On the Variance of the Adaptive Learning Rate and Beyond," which addresses some of the questions raised by the authors. Please discuss the unique contributions of your work compared to the variance-based analysis presented in that paper.

2. Although the authors try hard to explain the need for warmup and how to potentially reduce it, I still did not find persuasive answers to the questions posed. The conclusions are primarily based on intuitive narrative explanations and a simple experiment involving GPT-2. Meanwhile, some of the conclusions seem to be evident. For instance, before I read the paper, I could understand the statement "L2 update size is not sufficient to quantify the 'effectively' large updates". The paper lacks convincing evidence to support its claims. Lastly, I recommend that the authors narrow down the scope of the title to accurately reflect the content presented in the paper.

3. The authors use linear transitions to analyze the representation changes, which seems too toy for me.

4. I think the gradient clipping operation may be quite related regarding the authors' idea, as it directly impacts the adaptive LR. Could the authors provide some research here?

5. Regarding the writing of this paper, in my opinion, it is not particularly easy to follow. The organization feels somewhat messy. I think the authors should improve the clarity and structure. For example, including more detailed explanations and transitions between sections.

6. In Figure 1, I observe that the performance may be quite similar when using a lower learning rate. Could the authors specify the lowest learning rate used in your experiments?

7. I found the authors' use of the term "update size" to denote the step size in Adam somewhat confusing. I recommend that the authors use "update step", "adaptive learning rate" or "effective learning rate" instead, as these terms are clearer.

**Questions:**

See Weakness.

**Limitations:**

I have not found any discussions about the limitations and potential negative societal impact. But in my opinion, this may not be a problem, since the work only focuses on analyzing the warmup heuristic in machine learning. Still, it is highly encouraged to add corresponding discussions.

---

> ### Author Rebuttal · Authors · 2024-08-06
>
> Thank you for your review and feedback!
>
> **W7 Update Size:** We want to begin by clarifying this as it is a very important concept for our paper. By “update size” we literally mean “the size of the update”. This can be measured in different ways, for example through the L2 norm of the update, the angular change, or the representation change. We note that this is different from the learning rate, which scales the update size, but does generally not fully determine the update size. For example in SGD the L2 update size depends on the learning rate as well as the norm of the gradient. We specifically use “update size” or “update magnitude” to try to avoid confusion with these other terms that we do not feel capture our intended meaning.
>
> **W1 Comparison with RAdamW:** Thank you for this suggestion for improving our paper. Since RAdamW is a popular method for reducing the need for warmup it serves as a good comparison and contextualization of our work. Figure 1 in the global response shows that the RAdamW modifications are insufficient to eliminate the need for warmup in our setting. [2] suggest that RAdamW is approximately equivalent to 4 steps of SGDM followed by Adam with a special type of built-in warmup whose length is roughly 2/(1-beta2) = 40, which is likely too short in our setting.
>
> The analysis in [1] is based on the idea that early in training the second-moment estimates are not accurate (noisy) and can therefore not be trusted to scale the update properly. This could in turn contribute to the need for warmup. We note that without momentum, perfect estimates of the second moment at the current time step would control the expected L2 norm of the update. This relates our approach of looking at the update size to the adaptive learning rate view you seem to favor. We note that although [1] focuses on this issue of counteracting noisy estimates of the second moment, this is not necessary the sole reason warmup is beneficial. This is supported by the fact that both SGD and Lion empirically need warmup in various settings but do not use the second moment at all, indicating there is more to the story.
>
> The other aspects we explore relate to the momentum (e.g. the bias correction), weight decay and initialization (the angular updates), and how the gradient diversity affects how quickly the internal representations change (RRC compensation). We don’t believe [1] touches upon any of these to a significant extent. We also use Lion to control the L2 update size, avoiding the issues that [1] focuses on. Both works aim to reduce the need for learning rate warmup but focus on different aspects and take a completely different approach. Overall we therefore believe there is very little overlap in the contributions of these works, and if anything they could be complementary if we were to port our Lion modifications back to AdamW.
>
> **W2 Title:** Yes, we agree that just using GPT instead of neural networks would be more fitting. Thank you for this suggestion.
>
> **W2 L2 Norm is obvious:** We don’t believe this is the case, see both the discussion of gradient clipping and RAdamW which are closely related to this idea and not at all obvious shouldn’t work. We also note that clipping the L2 norm of the update will prevent a network from diverging to infinity, something that is often observed in unstable SGD training where warmup can help. Could you clarify further why you believe this is obvious or provide some reference to prior work that shows this?
>
> **W3 Linear Transformations:** This is of course a simplification compared to full neural networks. However, we believe this already gives interesting insights that are sufficient for our purposes. Specifically, this clearly shows that the parameter update size must be decreased when the gradient diversity is low in order to keep the representation changes small. We are also able to draw the same conclusions as existing works e.g. muP and hyperparameter scaling laws, showing this analysis provides useful results despite its simplicity. We believe a more complicated analysis would not fit within the scope of this paper, which is already on the broad side as you mention.
>
> **W4 Gradient Clipping:** Yes, gradient clipping can directly affect the update size. This is especially true when using SGD without momentum, where gradient clipping is similar to clipping the L2 norm of the update. However, controlling the L2 norm is not sufficient as we show. We believe the other ideas like the angular updates and representation changes are less related (but please clarify if you believe they are).
>
> **W5 Presentation:** Could you be more specific here? We note that some of the other reviewers found the paper to be “very well written” and “well motivated and systematic”, but we are happy to consider any concrete changes you recommend.
>
> **W6 Range of the sweep:** Yes, the performance is similar for smaller learning rates. This is consistent with our main hypothesis that large updates cause the degradation that warmup counteracts. Smaller learning rates result in smaller updates, thus avoiding this issue. The lowest learning rate in this sweep is 3e-4.
>
> Please let us know if you have any further questions or suggestions! If you feel we have at least partially addressed your concerns, we would greatly appreciate it if you would consider reevaluating your review score.
>
> ---
>
> [1] Liyuan Liu, Haoming Jiang, Pengcheng He, Weizhu Chen, Xiaodong Liu, Jianfeng Gao, and Jiawei Han. On the variance of the adaptive learning rate and beyond. In International Conference on Learning Representations, 2020. URL https://openreview.net/forum? id=rkgz2aEKDr. arXiv:1908.03265.
>
> [2] Jerry Ma and Denis Yarats. On the adequacy of untuned warmup for adaptive optimization. In Proceedings of the AAAI Conference on Artificial Intelligence, volume 35, pages 8828–8836, 2021. arXiv:1910.04209.

---

> > ### Comment · Reviewer_ramW · 2024-08-11
> >
> > Thanks for the response. I have no further concerns. Considering the authors' further revisions based on the rebuttal, I vote for acceptance.

---

### Official Review · Reviewer_gjgG · 2024-07-11

**Soundness:** 3
**Presentation:** 3
**Contribution:** 4
**Rating:** 6
**Confidence:** 4

**Summary:**

The submission analyzes the underlying reason behind the need for a learning rate warmup in neural network training, focusing on GPT pre-training with AdamW and Lion optimizers. The authors identify three key reasons as to why the initial updates are large:
1. Momentum handling by AdamW,
2. Early updates not correlated with the initial weight magnitudes
3. Correlation between gradients of examples during early training

The study introduces modifications to the Lion optimizer to mitigate the first two issues and proposes a method for controlling activation updates to address the third. Overall, I believe the paper's contributions are significant and hence I vote for acceptance.

**Strengths:**

* This paper analyzes various metrics that correlate with the benefit of learning rate.
* The analysis of the normalized Gradient Descent is insightful and reproduces previously known scaling laws.

**Weaknesses:**

* The authors begin by analyzing warmup for Adam and instead of directly modifying Adam, they modify the Lion optimizer. Direct modifications to the Adam optimizer would be more convincing and then moving to Lion would streamline the arguments.
* The LionAR algorithm is a much more complex solution than AdamW + warmup. Warmup duration is not a crucial hyperparameter, as a longer warmup duration does not hurt training.
* The experiments are performed on a fixed setup: GPT-2 model with 100M parameters trained on a single dataset. To ensure the validity and generalizability of the results, it is crucial to extend the analysis to various model architectures, parameter sizes, and datasets.

**Questions:**

* Which dataset is used for training the model? I don't think it is mentioned anywhere in the paper and its important for reproducing the results.
* Can the authors clarify the statement 'This factor is larger if the gradients are negatively correlated, which we empirically observe often happens early in training' on line 110?
* Did the authors try Adam with inverse bias correction for the momentum as suggested by equation 1?

**Limitations:**

See weaknesses

---

> ### Author Rebuttal · Authors · 2024-08-06
>
> Thank you for your review and feedback!
>
> **Using Adam instead of Lion:** We actually experimented with direct modification to Adam originally before moving to Lion. This worked equally well or better than the Lion modifications. However, we realized that at the start of training the gradient norms can change rapidly, specifically decreasing in our setup. This causes the second moment in Adam to be larger than it would otherwise be, resulting in smaller update sizes and essentially giving an additional warmup-like effect. The update size is also affected by the alignment of the gradients in successive steps (through momentum), complicating the control of the update size. The main reason for moving to Lion was that it gives precise control of the update size at a given iteration, eliminating these confounding effects that we found hard to control for in Adam and allowing us to explore the effect of the update size directly. We will edit the manuscript to clarify this, but we acknowledge your point that the use of Lion complicates the story.
>
> **Complexity of LionAR:** This is a fair point, the algorithm for LionAR is more complicated than for AdamW. A significant portion of this complexity comes from trying to transfer the hyperparameters of Adam over to Lion, as well as handling the two types of momentum. This could be removed if we don’t need this which would simplify the algorithm considerably. Then the major difference is additional projection of the weight norm instead of using weight decay. In practice weight decay is also only applied to some parameters which could be expressed with a similar if/else branch. In terms of the optimization dynamics we expect the behavior of LionAR to be much more regular than when using weight decay which could matter more than the complexity of the code overall. That being said, we also agree that warmup is a perfectly fine solution in practice, we primarily want to offer insights into why it is needed.
>
> **Diversity of the Experiments:** We agree and would have liked to showcase a broader range of experiments. We are unfortunately somewhat restricted in our compute budget but have tried to add some simple ablations in the global response. We tried a different dataset (SlimPajama [1]) with the same GPT setup and found similar results. We also tried to change the architecture to a llama2 instead of the GPT2 style we used originally. In this case LionAR already suffices to eliminate the need for warmup without further tricks like the RRC or momentum changes. This is likely due to changes in the critical batch size although the exact reasons for why the architecture affects this are not clear to us. We want to further experiment with larger batch sizes and see if the llama behavior becomes more similar to that of GPT2.
>
> **Dataset:** Thank you for pointing this out. This was an oversight on our part, and we appreciate your thorough review. The dataset we used is OpenWebText [2], the same one referenced in the GPT2 paper and NanoGPT. We will update the manuscript to include this information.
>
> **Line 110 clarification:** We meant correlation over time / between optimizer steps, i.e. that when the gradients of successive steps point in opposite directions they cancel out in the momentum vector. This leads to smaller update sizes in Adam. However, the bias correction assumes that all future gradients will align with the current momentum vector and magnifies the update size accordingly, leading to much larger steps than we would see otherwise. We will edit this to clarify.
>
> **Adam with inverse bias correction** Yes, we did some exploratory experiments with modifications like these as well as removing the bias correction completely. This helps a bit, but overall the effect is too small to make a significant difference for momentum values like 0.9. At best this could result in warmup like effects of maybe 20 steps, which is too small to significantly decrease the no-warmup degradation in our setting. For comparison, see also the RAdamW results in the global response which might give effects similar to 40 steps of warmup which is not sufficient.
>
> ---
>
> [1]: Soboleva, Daria, Faisal Al-Khateeb, Robert Myers, Jacob R. Steeves, Joel Hestness, and Nolan Dey. "SlimPajama: A 627B Token Cleaned and Deduplicated Version of RedPajama." June 2023, www.cerebras.net/blog/slimpajama-a-627b-token-cleaned-and-deduplicated-version-of-redpajama.
>
> [2]: Gokaslan, Aaron, and Vanya Cohen. "OpenWebText Corpus." 2019, Skylion007.github.io/OpenWebTextCorpus.

---

> ### Comment · Reviewer_gjgG · 2024-08-08
>
> I thank the reviewers for their comments. Most of my concerns have been resolved. I look forward to the updated version of the manuscript.

---

### Official Review · Reviewer_pDv7 · 2024-07-12

**Soundness:** 3
**Presentation:** 3
**Contribution:** 2
**Rating:** 6
**Confidence:** 3

**Summary:**

In this paper the authors investigate the performance benefits seen from the common practice of learning rate warmup and scheduling, attempt to understand the mechanistic underpinnings of those improvements, and engineer optimizers that mitigate the need for warmup. They conduct experiments using NanoGPT and consider controlling parameter updates, angular updates, and “relative representation” changes to close the gap between warmup and no warmup.

**Strengths:**

Warmup length and peak learning rate are certainly some of the most important hyperparameters in large model training, and eliminating the need for a warmup phase would present a significant simplification to training. The paper is well motivated and systematic in its investigation of warmup and proposals to sidestep the necessity of warmup. The RRC is an interesting and promising angle on this question.

**Weaknesses:**

The results do not suggest a clear prescription for learning rate scaling or straightforward changes that can be made to initializations or updates. In particular, the RRC is completely dependent on the inputs, but there does not seem to be any discussion or investigation of the effects of the input data. For the NanoGPT experiments there is no mention of what the data is. Presumably training was done with a cross entropy loss, but this is also not mentioned.

**Questions:**

Can the authors provide some discussion about the sensitivity of RRC to variance in the inputs across and within batches? It is also not clear what direction the RRC results are suggesting. An “automatic warmup” that scales the update sizes according to online measurements of the signal-to-noise ratio is still a warmup phase, albeit a more principled way to arrive at what that schedule should look like. To be clear, I don’t think this is a bad thing, but it may be more representative of the results to propose an “adaptive” or “automatic” learning rate scheduler, rather than claim to make progress towards eliminating the need for scheduling.

**Limitations:**

Partially

---

> ### Author Rebuttal · Authors · 2024-08-06
>
> Thank you for your review and feedback!
>
> **Missing Dataset Information:** Thank you for pointing this out! This was an unfortunate oversight on our behalf (and we geatly appreciate your attention to these details). The dataset we train on is the original OpenWebText dataset [1] used in the original GPT2 work. Training is indeed performed via next token prediction using cross entropy with teacher forcing. We will include this in future revisions of the manuscript.
>
> **RRC Dependency on Data:** We have expanded our experiments to include a separate dataset (SlimPajama [2]), obtaining similar results. You are correct that the RRC correction depends on the input data (by design), in a way that controlling the parameter update norm can not. This will vary depending on the data within a batch. The simplest example of this would be if the whole mini-batch consists of the same datapoint repeated. The RRC correction would indicate that we should use a lower learning rate (parameter update size) in this case than if the data has no similarity.
>
> However, the RRC correction does not measure the similarity of the data in the input space but rather through the gradient diversity. It therefore not only depends on the similarity of the data within each batch (which shouldn’t vary throughout training) but rather on the similarity of the “learning” to be made from each sequence. This can vary throughout time, for example early in training the model could largely learn syntax, word frequency, and potentially unlearn initialization biases (like maybe outputting the input sample). There is a strong overlap in the “lesson” to be learned from each input sequence in this context, resulting in similarity between the gradients which may be dominated by these simple concepts. Later in training the model could learn more advanced concepts that differ more depending on the semantic content of each sequence, resulting in greater gradient diversity. At this point we can perform larger parameter updates because the contributions of the input sequences do not all line up to change the representations in the same way. Such alignment would lead to large representation changes, which we hypothesized could lead to lasting issues e.g. with the non-linearities.
>
> In its current form there is no dependency on the similarity between batches for the RRC correction. We hope that if the data is well shuffled the similarity within each mini-batch also reflects the similarity between batches, otherwise this could lead to issues. Let us know if you believe this kind of conceptual discussion of the RRC would be useful / informative we would be happy to incorporate it into the manuscript.
>
> **Eliminating Warmup vs Automatic Warmup:** This is a fair point and we do indeed refer to the RRC correction as an automatic warmup in lines 41 and 220. Overall we believe the definition of a warmup or a schedule is a bit subjective. For example Adam can be seen as a per-coordinate adaptive scheduler for the vanilla SGD learning rate, but what we refer to as the learning rate schedule does typically not account for this. In the same way an RRC corrected optimizer could eliminate the need for manually specified warmup, but would lead to a warmup like effect in the parameter update size. Using other metrics to measure the update size like the RRC, there would not necessarily be an observable warmup phase (and the learning rate could directly control this update size).
>
> **Clear Recommendations:** We do not touch upon initialization but we do show optimizer modifications (LionAR) and potentially further scaling via the RRC gradient noise correction is sufficient to significantly reduce or eliminate the need for warmup. People could attempt to directly apply these methods in other settings if warmup is undesirable for some reason. However we view the primary contribution of our work to be understanding of how the changes in the update size across time contribute to the need for warmup. We hope this will lead to a better understanding of optimization dynamics, that it could inform practitioners about the length of warmup (by just measuring the same metrics we do), and finally lead to better optimizer design in the future. We take the steps towards the optimizer design with LionAR and the RRC correction, but believe future work could improve upon them.
>
> **Additional Experiments:** Aside from repeating our experiments with a different dataset we have performed several other ablations for additional experimental evidence, see the global response.
>
> Please let us know if you have any additional questions! We would also greatly appreciate it if you could inform us whether the additional experiments, proposed modifications, and clarifications at least partially mitigate your concerns.
>
> ---
>
> [1]: Gokaslan, Aaron, and Vanya Cohen. "OpenWebText Corpus." 2019, Skylion007.github.io/OpenWebTextCorpus.
>
> [2]: Soboleva, Daria, Faisal Al-Khateeb, Robert Myers, Jacob R. Steeves, Joel Hestness, and Nolan Dey. "SlimPajama: A 627B Token Cleaned and Deduplicated Version of RedPajama." June 2023, www.cerebras.net/blog/slimpajama-a-627b-token-cleaned-and-deduplicated-version-of-redpajama.

---

> > ### Comment · Reviewer_pDv7 · 2024-08-12
> >
> > I thank the authors for their detailed response. They have addressed the points raised in my review and I have increased my score.

---

### Official Review · Reviewer_VUaC · 2024-07-26

**Soundness:** 3
**Presentation:** 4
**Contribution:** 3
**Rating:** 7
**Confidence:** 2

**Summary:**

To train current deep neural network architectures, especially transformers, the learning rate of AdamW is usually first linearly increased to reach a peak before it's decreased to zero. The paper analyzes the impact of this so-called warming-up phase on GPT-2 models from the perspective of the update size. As a second contribution, the paper presents some small modifications for the Lion optimizer to mitigate some of the issues encountered in the experiments.

**Strengths:**

- The paper is very well written. The experiments are nicely motivated and reasonable.

- Warming up the learning rate is arguably common practice for training transformer models, but not well understood. The paper provides some interesting analysis of the matter, which could potentially lead to a more intuitive understanding of the problem and eventually better optimizers.

**Weaknesses:**

- The results of the paper are somewhat inconclusive and after reading the paper, I am still not sure about the dynamics during the warming up phase. For example, while controlling angular updates seems to stabilize training to a certain degree, it eventually doesn't lead to better performance. Also, as the paper clearly states, the magnitude of the parameter updates doesn't really account for the gains of the warm-up phase. I am wondering if the paper approaches the problem actually from the right perspective. Having said that, I think the paper still provides some value and might help to stir future research.

- While the empirical evaluation is insightful, it's limited to a single architecture and domain. This raises the question of how reliable the results actually are.

**Questions:**

- How sensitive are the results from Section 3 to the type of learning rate schedule? For example, how would Figure 1 look if you used, let's say, a cosine annealing schedule?

**Limitations:**

I think the paper spells out all its limitations; however, for visibility, it might be better to move the corresponding paragraph from the appendix to the main text. I don't see any negative societal impacts of the work.

---

> ### Author Rebuttal · Authors · 2024-08-06
>
> Thank you for your review and feedback!
>
> **Generalizability of the results:** We have added experiments with a different GPT architecture (Llama2) and dataset (SlimPajama), see global response. We find that the results are similar, suggesting some transferability within GPT-style training. Originally we wanted to include more transformer tasks like vision transformers and translation but actually found that warmup did not have a significant impact in our target settings. This likely depends on the batch size among other factors, but instead of exploring this further we decided to narrow the scope to GPTs. We will change the title to reflect this, changing “neural network” to “GPT” as suggested by one of the other reviewers.
>
> **General Approach:** The main takeaway would be that the need for warmup largely arises from poor control of the update size during training. Modified optimizers, especially those that control the angular update size directly, can significantly reduce or eliminate the need. However, ultimately we believe that it is large changes in the internal representations that cause the need for warmup, which can not be fully captured by simple measures of the update size of the parameters. We show that the discrepancy between the parameter update size and representation changes can be linked to the noise in the gradient. We can compensate for this based on measurements of the gradient noise, leading to something like an
> “automatic warmup” in the parameter update size. The hope is that this will lead to an improved understanding of optimization dynamics and potentially eventual improvements in optimizer design. Controlling the angular updates improves performance without warmup, but we did not really attempt to improve the overall performance with warmup. We note that LionAR is at a bit of a disadvantage since the weight decay value and other settings are inherited from the baseline (not re-tuned) and we constrain the magnitude without any additional mechanism like learnable gains to compensate for this.
>
> **Cosine Schedule:** Thank you for bringing this up, this is something we will clarify further. We reran the AdamW baseline with a cosine schedule and added it to the global response (see left half of top figure). The trapezoidal schedule (aka warmup-stable-decay) we used in the manuscript has been becoming more popular for LLM training since it provides more flexibility in the training duration and can modify the data mixture in the cooldown phase while giving similar results [1, 2]. The reason we opted to use it is that it clearly separates the warmup phase from the rest of training, unlike common variants of the cosine schedule where the length of the warmup simultaneously affects the shape of the rest of the schedule. We wanted to eliminate this as a confounding factor, so that the apparent benefits of warmup were not coming from changes in the cooldown phase. **In terms of the gap between warmup and no warmup, the results are very similar.** However, in this case the cosine schedule performs marginally better and the learning rate transfers better between different warmup lengths. We believe the latter effect is because the area under the curve is roughly independent of the warmup length for the cosine (because the whole schedule shifts), but not the trapezoidal schedule.
>
> **Limitations:** Yes this is a fair point, we will move the limitations section to the main body in a future version of the manuscript.
>
> Please let us know if you have any additional questions! We would also greatly appreciate it if you could inform us whether the additional experiments, proposed modifications, and clarifications at least partially mitigate your concerns.
>
> ---
>
> [1]: Hu, Shengding, et al. "Minicpm: Unveiling the potential of small language models with scalable training strategies." arXiv preprint arXiv:2404.06395 (2024).
>
> [2]: Hägele, Alexander, et al. "Scaling Laws and Compute-Optimal Training Beyond Fixed Training Durations." arXiv preprint arXiv:2405.18392 (2024).

---

> > ### Comment · Reviewer_VUaC · 2024-08-12
> > **reply to authors**
> >
> > I thank the authors for addressing my comments. I will raise my score and vote for acceptance of the paper

---

### Author Rebuttal · Authors · 2024-08-06

We would like to thank all the reviewers for their thoughtful reviews and feedback on our manuscript. We will try to address your specific concerns and questions in our individual responses.

Here we present additional experimental results with accompanying plots in the pdf:
* Figure 1 shows what the baseline plots would look like for a **cosine decay schedule** (specifically one-cycle cosine without momentum for these experiments). The reason we went with the trapezoidal schedule is that it fully separates the warmup phase from the rest of the schedule unlike in the cosine schedule where the warmup length affects the whole schedule. We wanted to avoid these confounding effects with our choice of the trapezoidal schedule.
* Figure 1 also shows the performance of **RAdamW** [1], a popular optimizer modification for reducing the need for warmup, in the baseline setup. We find that while it helps, it does not eliminate the need for warmup. The analysis of [2] suggest RAdamW functions similar to a 2/(1-beta2)=40 step warmup which seems to roughly match our findings (2% here would be 100 steps).
* Figure 2 shows the **effects of changing the dataset used in our experiments** from OpenWebText [3] to SlimPajama [4]. Overall the results are similar as before, when controlling the L2 norm via LionA warmup is still beneficial, controlling the angular updates via LionAR decreases the gap significantly. The higher momentum LionAR with Nesterov momentum and our momentum correction eliminates the gap fully. The RRC also seems to eliminate the benefit of warmup but still has the same practical limitations as we describe in section 6.1.
* Figure 3 shows the **effects of changing the architecture from GPT2 to the llama style [5]** while keeping the dataset and parameter count (~124m) the same. This includes using SwiGLU activations, RoPE embeddings and RMSNorm. In this case LionAR is able to fully eliminate the need for warmup without any additional tricks like the RRC compensation or momentum corrections. Based on our analysis these additional tricks are likely only needed when the critical batch size is very small initially. In the future we want to rerun these experiments using a larger batch size to verify this, but were not able to do it in time for this rebuttal.
* Figure 4 shows the results for a **larger 209m parameter llama2** trained on SlimPajama. Overall the results are similar to the smaller llama.

**We believe these additional experiments can increase the variety of our experimental setup, helping mitigate this limitation somewhat**, although an even broader range would of course be preferable.

Based on reviewer feedback, we have decided to limit our experimental scope to GPT variants and will change the title of the paper to reflect this, as suggested by one of the reviewers. Previously we wanted to include broader transformer experiments like DeiT and translation but found it hard to identify good setups where warmup has a significant impact but are computationally tractable for us. We found that many of the reference configurations we experimented with use warmup but don’t actually benefit significantly from it. This likely varies across batch sizes and other configuration aspects we found too expensive to tune.

---

[1] Liyuan Liu, Haoming Jiang, Pengcheng He, Weizhu Chen, Xiaodong Liu, Jianfeng Gao, and Jiawei Han. On the variance of the adaptive learning rate and beyond. In International Conference on Learning Representations, 2020. URL https://openreview.net/forum? id=rkgz2aEKDr. arXiv:1908.03265.

[2] Jerry Ma and Denis Yarats. On the adequacy of untuned warmup for adaptive optimization. In Proceedings of the AAAI Conference on Artificial Intelligence, volume 35, pages 8828–8836, 2021. arXiv:1910.04209.

[3]: Gokaslan, Aaron, and Vanya Cohen. "OpenWebText Corpus." 2019, Skylion007.github.io/OpenWebTextCorpus.

[4]: Soboleva, Daria, Faisal Al-Khateeb, Robert Myers, Jacob R. Steeves, Joel Hestness, and Nolan Dey. "SlimPajama: A 627B Token Cleaned and Deduplicated Version of RedPajama." June 2023, www.cerebras.net/blog/slimpajama-a-627b-token-cleaned-and-deduplicated-version-of-redpajama.

[5]: Touvron, Hugo, et al. "Llama 2: Open foundation and fine-tuned chat models." arXiv preprint arXiv:2307.09288 (2023).

---

### Decision · Program_Chairs · 2024-09-25

**Decision:**

Accept (poster)

**Comment:**

The paper under consideration investigates the common but not well-understood practice of warming up learning rates while training transformer models.
Reviewers appreciated the importance of the topic and the clarity of the writing. In addition, reviewers expressed appreciation for the systematic investigation put forth in the paper.
While the reviewers noted some limitations, including the inconclusiveness of the results, their primary concern was the study's initial narrow focus on a single architecture and dataset. However, the authors' post-rebuttal additions, which included additional experiments, successfully addressed this concern by providing consistent results that reinforced the initial findings.
Overall, reviewers were overwhelmingly positive in their assessment and all unanimously recommended acceptance of the paper.